# Acetylation-induced TDP-43 pathology is suppressed by an HSF1-dependent chaperone program

Ping Wang[1], Connor M. Wander[1], Chao-Xing Yuan[2], Michael S. Bereman[3] & Todd J. Cohen[1]

TDP-43 pathology marks a spectrum of multisystem proteinopathies including amyotrophic lateral sclerosis, frontotemporal lobar degeneration, and sporadic inclusion body myositis. Surprisingly, it has been challenging to recapitulate this pathology, highlighting an incomplete understanding of TDP-43 regulatory mechanisms. Here we provide evidence supporting TDP-43 acetylation as a trigger for disease pathology. Using cultured cells and mouse skeletal muscle, we show that TDP-43 acetylation-mimics promote TDP-43 phosphorylation and ubiquitination, perturb mitochondria, and initiate degenerative inflammatory responses that resemble sporadic inclusion body myositis pathology. Analysis of functionally linked amyotrophic lateral sclerosis proteins revealed recruitment of p62, ubiquilin-2, and optineurin to TDP-43 aggregates. We demonstrate that TDP-43 acetylation-mimic pathology is potently suppressed by an HSF1-dependent mechanism that disaggregates TDP-43. Our study illustrates bidirectional TDP-43 processing in which TDP-43 aggregation is targeted by a coordinated chaperone response. Thus, activation or restoration of refolding mechanisms may alleviate TDP-43 aggregation in tissues that are uniquely susceptible to TDP-43 proteinopathies.

[1] Department of Neurology, UNC Neuroscience Center, University of North Carolina, Chapel Hill, NC 27599, USA. [2] Alexion Pharmaceuticals Inc, New Haven, CT 06510, USA. [3] Department of Biological Sciences and Department of Chemistry, Center for Human Health and the Environment, North Carolina State University, Raleigh, NC 27695, USA. Correspondence and requests for materials should be addressed to T.J.C. (email: toddcohen@neurology.unc.edu)

TDP-43 is the dominant pathology identified in most amyotrophic lateral sclerosis (ALS) and ~50% of frontotemporal lobar degeneration (FTLD-TDP) patients. As an RNA-binding protein, TDP-43 possesses two RNA-recognition motifs (RRMs), and a C-terminal prion-like domain that harbors the majority of the familial ALS-associated mutations[1–5]. While TDP-43 is predominantly nuclear localized under normal conditions, pathological TDP-43 found in diseased brain and spinal cord is abnormally aggregated primarily in the cytoplasm, which correlates with the onset and progression of TDP-43 proteinopathy by several distinct pathogenic mechanisms[6–11].

In addition to the central nervous system (CNS), TDP-43 also forms pathological inclusions in skeletal muscles of patients with sporadic inclusion body myositis (sIBM)[12–16], suggesting TDP-43 pathology is not solely limited to vulnerable neurons, but likely induces multisystem proteinopathy in susceptible cell types.

Although it is not fully understood how TDP-43 causes cellular toxicity, substantial evidence indicates that TDP-43 aggregates induce loss of normal nuclear TDP-43 functions, as nuclear depletion of normally soluble TDP-43 and sequestration into inclusions leads to reduced splicing and RNA stability as well as activation of cryptic splice sites in cultured cells and transgenic

**Fig. 1** TDP-43 acetylation-mimics promote TDP-43 aggregation in cells. **a** QBI-293 cells expressing TDP-43-ΔNLS in the absence of ectopic acetyltransferases were analyzed by mass spectrometry, which identified the K145-acetylated peptide. **b** A schematic diagram of GFP-tagged TDP-43 constructs used in this study. Basic residues (*light brown*) within the bipartite nuclear localization sequence (NLS) were mutated to generate cytoplasmic TDP-43 (TDP-43-ΔNLS). A single glutamine was substituted at position K145 to generate the acetylation-mimic K145Q. **c** Cells expressing wild-type TDP-43 and TDP-43-K145Q or cytoplasmic localized TDP-43-ΔNLS and TDP-43-ΔNLS-K145Q were analyzed by double-labeling to detect TDP-43 (*green*), phospho-TDP-43 using P409/410 (*red*), and DAPI (blue) to detect nuclei. *White arrows* highlight highly phosphorylated TDP-43 aggregates. Scale bar = 50 μm. **d** Soluble and insoluble fractions isolated from cells expressing the indicated TDP-43 constructs were analyzed by immunoblotting using TDP-43, P409/410, or GAPDH antibodies. Shown is a representative image from $N = 3$ independent experiments. The asterisk indicates the accumulation of ~75–250 kDa TDP-43 species. **e** Quantification of TDP-43 and phospho-TDP-43 levels illustrate increased insoluble levels of TDP-43-ΔNLS-K145Q. Error bars indicate SEM and the asterisks indicates statistical significance with ***$p$-value < 0.001, **$p$-value < 0.01, and *$p$-value < 0.05, as measured by Student's $t$-test from $N = 3$ biological replicates

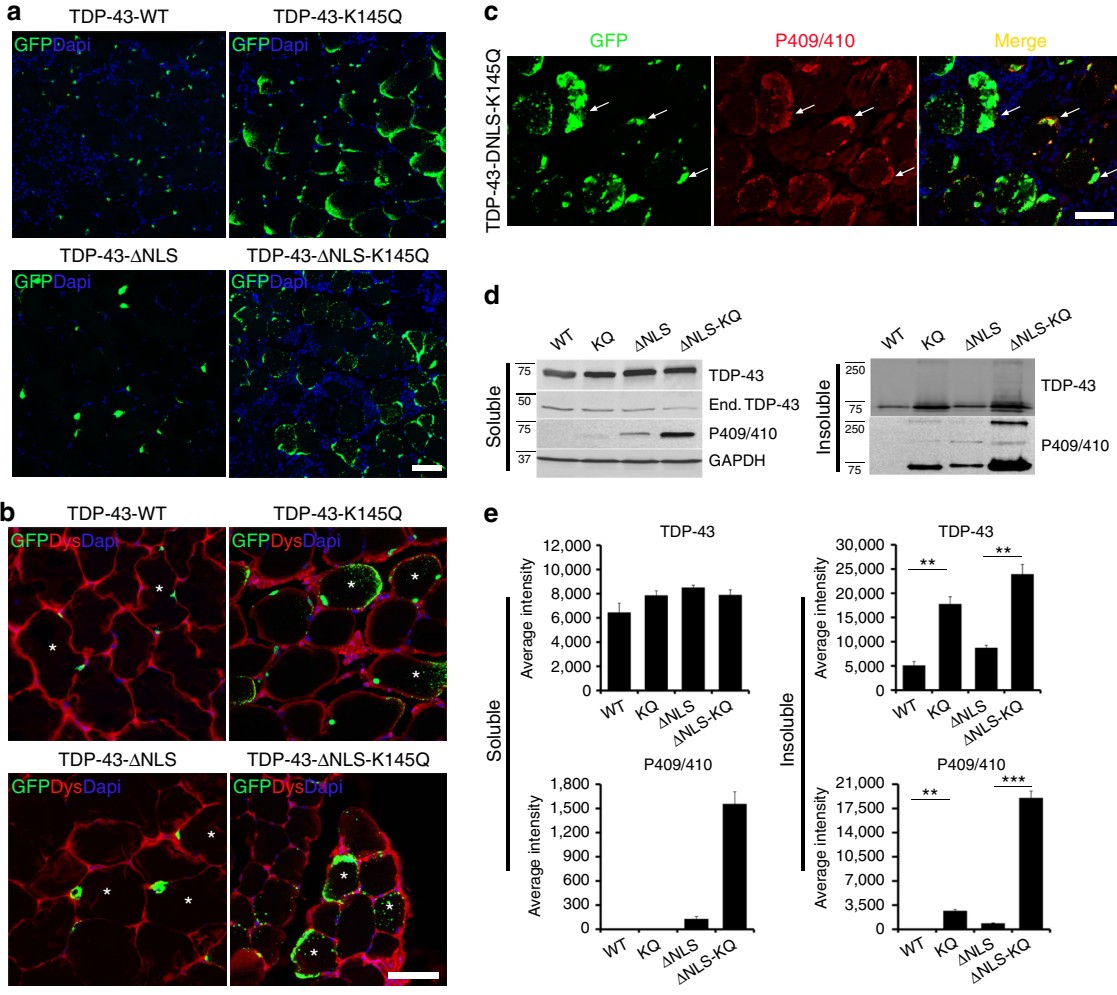

**Fig. 2** TDP-43 acetylation-mimics generate hallmark disease pathology in vivo in mouse skeletal muscle. **a** Immunofluorescence analysis was performed on mouse tibialis anterior (TA) muscles electroporated with the indicated GFP-tagged TDP-43 plasmids for 4 days. **b** TA muscle sections were double-labeled with GFP (*green*) and dystrophin (Dys, *red*) antibodies to mark the muscle periphery, while nuclei were stained with DAPI (*blue*). **c** TA muscles expressing TDP-43-ΔNLS-K145Q were double-labeled with GFP and phospho-TDP-43 (P409/410) antibodies to mark pathological TDP-43 aggregates. *White arrows* highlight phosphorylated TDP-43 accumulation. **d** Electroporated TA muscles were harvested, fractionated into soluble and insoluble fractions, and analyzed by immunoblotting using TDP-43, P409/410, and GAPDH antibodies. Shown is a representative immunoblot from $N = 3$ independent experiments. **e** Quantification of **d** illustrates increased insoluble TDP-43 in muscles expressing TDP-43-ΔNLS-K145Q. Error bars indicate SEM, and the asterisk indicates statistical significance with ***$p$-value < 0.001 and **$p$-value < 0.01, as measured by Student's $t$-test. Scale bar = 50 μm

mice[17–19]. As a compensatory response to loss of TDP-43 function, auto-regulation of the *TARDBP* transcript increases TDP-43 levels[20–22], thereby generating a feed-forward pathogenic cycle consisting of sustained accumulation of cytosolic TDP-43 that further compromises normal nuclear TDP-43 functions and potentially alters expression of ~6000 target genes, as shown by genome-wide TDP-43-binding studies[19, 23, 24].

The significance of TDP-43 post-translational modifications as it relates to TDP-43 aggregation, stability, and clearance mechanisms is not fully understood. While phosphorylated TDP-43 at Ser-403/404 and Ser-409/410 is an excellent marker of disease pathology, several studies indicated that phosphorylation may prevent rather than promote TDP-43 aggregation[25, 26], suggesting that additional regulation could modulate TDP-43 function. We previously demonstrated that TDP-43 is subject to reversible lysine acetylation within the RRM domains (residues K145 and K192), which abrogates the ability of TDP-43 to bind RNA and also promotes the accumulation of TDP-43 aggregates in the nucleus and cytoplasm of cultured cells[27]. We proposed that acetylation-induced loss of function represents a new

pathogenic mechanism in tissues harboring mostly full-length, but not truncated, TDP-43 inclusions (e.g., ALS motor neurons[28]).

Although acetylated TDP-43 was detected in ALS patient spinal cord, the pathophysiological relevance of this modification remained uncertain. However, site-specific TDP-43 acetylation within the RRM domains combined with the close juxtaposition of the identified lysine residues (K145 and K192) to the TDP-43/RNA interface indicates that acetylation may alter charge-mediated interactions with nucleic acids[29, 30]. Thus, prolonged cytoplasmic accumulation could lead to TDP-43 acetylation and impaired recycling of cytoplasmic TDP-43 back to the nucleus. Any evidence that acetylation is sufficient to induce TDP-43 pathology in vivo has not been investigated but could provide valuable insight into the regulation of TDP-43 function and hence modulation of its downstream targets.

Chaperone-dependent refolding acts in a compensatory manner to suppress toxic aggregates in a variety of neurodegenerative diseases[31, 32]. In particular, HSF1 is thought to coordinate stress-activated gene transcription leading to chaperone

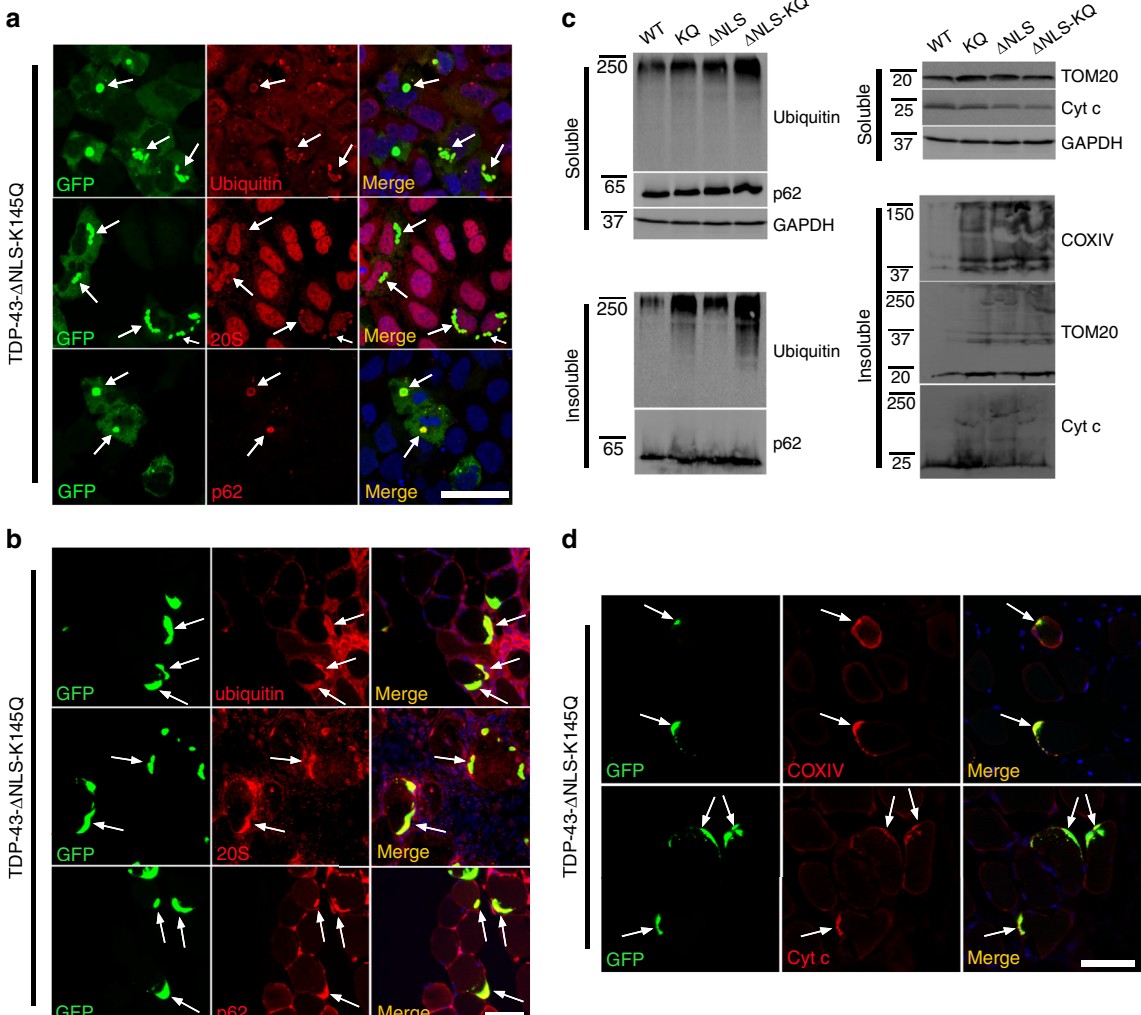

**Fig. 3** TDP-43 acetylation-mimic aggregates recruit the degradative machinery and promote mitochondrial dysfunction. **a** Cells expressing aggregate-prone TDP-43-ΔNLS-K145Q were analyzed by double-labeling using ubiquitin, 20S proteasome, or p62 antibodies, and counterstained with DAPI (*blue*). *White arrows* highlight co-localization of ubiquitin, 20S proteasome, or p62 with TDP-43 aggregates. **b** Mouse TA muscles electroporated with TDP-43-ΔNLS-K145Q for 14 days were similarly analyzed using ubiquitin, 20S proteasome, and p62 antibodies. *White arrows* highlight prominent co-localization of TDP-43 aggregates with ubiquitin, 20S proteasome, or p62 in mouse muscles. **c**, **d** Mouse tissues were analyzed by immunoblotting (**c**) or microscopy (**d**) using ubiquitin or p62 antibodies, as well as the mitochondrial panel of antibody markers (COXIV, TOM20 and Cytochrome c). Scale bar = 50 μm

induction, aggregate refolding, and re-establishment of cellular proteostasis[33]. Loss of HSF1 exacerbates neurodegeneration in several models[34, 35], while overexpression of HSF1 increases neuronal survival[36, 37]. Global genome-wide studies indicate that both small and large heat shock proteins (HSPs) as well as non-HSP targets may be critical for HSF1-dependent neuroprotection[38, 39]. However, HSF1 activation may not be readily achieved in neurons due to a higher threshold requirement for sustained HSF1 activity[40, 41]. Nonetheless, the HSF-1 targets Hsp40 and Hsp70 were recruited to full-length TDP-43 foci and phosphorylated C-terminal fragments[42] and also were sufficient to partially suppress TDP-43 aggregation[43, 44]. Whether increased chaperone activity is a viable strategy to alleviate aggregate burden in TDP-43 proteinopathies is not clear.

Here, we show that expression of acetylation-mimic TDP-43 mutants at residue K145 induces robust TDP-43 pathology characterized by insoluble, hyper-phosphorylated, and ubiquitinated TDP-43 aggregates that are readily engaged by components of the autophagy and ubiquitin-proteasome degradation machinery. To alleviate acetylation mimic-induced aggregation, HSF1 was activated by genetic or pharmacological approaches,

which led to induction of a transcriptional chaperone program capable of clearing cytoplasmic TDP-43 aggregates. Our study provides bidirectional formation and clearance of TDP-43 pathology, the latter of which occurs via an endogenous refolding program, suggesting that latent chaperone activity holds promise for the amelioration of TDP-43 pathology in the CNS and other susceptible tissue types.

## Results

**TDP-43 acetylation-mimic at K145 induces TDP-43 aggregation.** Our previous study identified acetylated residue K145 in RRM1 as a major site of TDP-43 acetylation induced by the acetyltransferase CREB-binding protein (CBP)[27]. Mass spectrometry analysis in the absence of ectopic CBP also detected K145 acetylation (Fig. 1a), suggesting steady-state TDP-43 acetylation occurs under normal cellular conditions. We therefore generated a series of GFP-tagged constructs expressing nuclear or cytoplasmic localized (ΔNLS) TDP-43 lacking or containing acetylation-mimic K-to-Q substitutions at position K145 (Fig. 1b). Compared to wild-type TDP-43 (TDP-43-WT), expression of nuclear acetylation-mimic TDP-43-K145Q formed

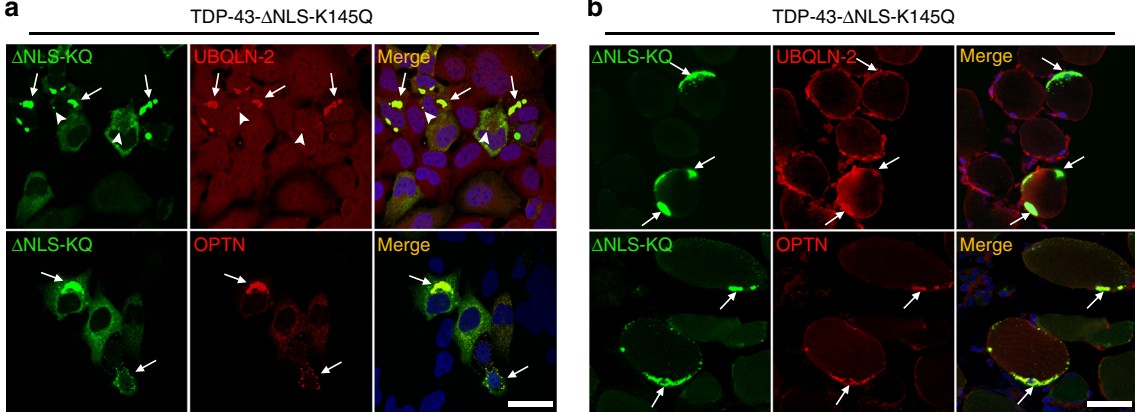

**Fig. 4** Functionally linked co-pathologies emerge in response to TDP-43 acetylation-mimic aggregates. QBI-293 cells (**a**) or TA muscles (**b**) were transfected with GFP-tagged cytoplasmic localized TDP-43-ΔNLS-K145Q either alone (endogenous UBQLN-2 analysis) or in tandem with an OPTN expression plasmid for increased detection sensitivity. Either endogenous UBQLN-2 or ectopically expressed OPTN were detected by double-labeling immunofluorescence using GFP (*green*), UBQLN-2 (*red*), or OPTN (*red*) antibodies, while DAPI marked nuclei (*blue*). We note significant recruitment of both UBQLN-2 and OPTN to TDP-43 aggregates in either cell-based or muscle fiber models. *White arrows* highlight recruitment of UBQLN-2 or OPTN to TDP-43 aggregates. *Arrowheads* indicate occasional co-localization of UBQLN-2 to smaller TDP-43 pre-aggregates. Scale bar = 50 μm

distinct stippled nuclear foci that were partially phosphorylated at known pathological epitopes (S409/410) (Fig. 1c, compare TDP-43-WT to TDP-43-K145Q). The cytoplasmic acetylation-mimic TDP-43-ΔNLS-K145Q produced inclusions that were more robustly phospho-TDP-43 immunoreactive (Fig. 1c, see TDP-43-ΔNLS-K145Q), resembling full-length cytoplasmic TDP-43 pathology in ALS spinal cord. In agreement with the immunofluorescence results, immunoblotting detected abundant insoluble and hyper-phosphorylated TDP-43-ΔNLS-K145Q aggregates (Fig. 1d, e, Supplementary Fig. 1). The accumulation of ~75–250 kDa TDP-43 species (Fig. 1d, see *asterisk*) is consistent with multimeric TDP-43 species observed in response to oxidative stressors[45]. In contrast to K-to-Q mutations, either K-to-R non-mimic mutations or known ALS-associated mutations, whether targeted to either the nucleus or the cytoplasm of cultured cells, has not produced comparable levels of TDP-43 aggregates[45]. Enhanced TDP-43 aggregation due to the K145Q acetylation-mimic was observed in all cell lines tested including motor neuron-like NSC-34 cells (Supplementary Fig. 2).

We hypothesized that TDP-43 aggregation would be conserved in both skeletal muscle and CNS tissues, both of which are vulnerable to TDP-43 pathology. We employed an in vivo electroporation system to efficiently deliver TDP-43 to mouse tibialis anterior (TA) muscles (Fig. 2a)[46]. Confocal imaging of electroporated muscle fibers showed nuclear localization of TDP-43-WT, while TDP-43-K145Q accumulated as punctate aggregates throughout the myofiber in both the nucleus and sarcoplasm. The TDP-43-K145Q localization pattern could indicate that myofibers generate a more cytoplasmic and aggregate-prone phenotype compared to that previously observed in cell lines[27]. Even more pronounced punctate aggregation was observed with the cytoplasmic localized TDP-43-ΔNLS-K145Q mutant (Fig. 2b, Supplementary Fig. 3, see aggregates within myofibers highlighted by *asterisk*). The ectopically produced mutant TDP-43 pathology generated in myofibers was robustly phosphorylated at S409/410 and showed abnormal localization to perinuclear regions, the muscle membrane, and also centrally within muscle fibers (Fig. 2c, *white arrows*, Supplementary Fig. 3). While TDP-43-WT displayed normal nuclear TDP-43 staining patterns, TDP-43-ΔNLS lacking acetylation-mimic mutations appeared to accumulate immediately surrounding and possibly adhered to the nuclear membrane (Fig. 2b, Supplementary Fig. 3). This unique TDP-43-ΔNLS staining pattern could similarly

reflect increased aggregation propensity in myofibers compared to cultured cells, leading to more rapid cytoplasmic deposition adjacent to the nuclear membrane. Subsequent immunoblotting of muscle homogenates confirmed the accumulation of phosphorylated TDP-43 aggregates generated by TDP-43-K145Q and more prominently by TDP-43-ΔNLS-K145Q (Fig. 2d, e, Supplementary Fig. 4). Thus, in vivo expression of TDP-43 acetylation-mimics is sufficient to rapidly (within days) generate TDP-43 pathology that resembles human sIBM skeletal muscle.

**TDP-43 acetylation-mimic alters cellular proteostasis.** To further characterize acetylation-mimic induced TDP-43 pathology, we analyzed known proteasome and autophagy markers implicated in TDP-43 protein clearance pathways[1]. In cells (Fig. 3a) or myofibers (Fig. 3b), a subset of cytoplasmic aggregates were ubiquitinated, as assessed by anti-ubiquitin antibodies, while the 20S proteasome was also recruited but was less prominently localized to TDP-43 aggregates (Fig. 3a, b, *white arrows*). Similarly, the autophagy marker p62 strongly localized to TDP-43 aggregates (Fig. 3a, b, *white arrows*). Consistent with impaired degradative function, however, we observed accumulation of high-molecular weight ubiquitin-positive smearing in insoluble muscle fractions (Fig. 3c, Supplementary Fig. 5). Indeed, mass spectrometry analysis of immunopurified TDP-43-ΔNLS-K145Q protein identified a specific ubiquitinated residue, K181, within the TDP-43 RRM linker region (Supplementary Fig. 6), further suggesting proteasome targeting of the aggregate-prone TDP-43 acetylation-mimic.

Mitochondria are perturbed in ALS patients and mouse models of ALS[47–52]. We therefore asked whether TDP-43 expressing muscle fibers displayed mitochondrial abnormalities in biochemical fractions of muscle extracts. While TDP-43-WT did not alter the levels of any mitochondrial markers tested, TDP-43-K145Q or cytoplasmic targeted TDP-43 (independent of its acetylation-mimic status) appeared to increase mitochondrial aggregation, as detected by high-molecular-weight species using three independent mitochondrial markers (TOM20, COXIV, and Cytochrome c) (Fig. 3c, Supplementary Fig. 5, see insoluble mitochondrial smearing[53]). There were negligible effects on soluble mitochondria contents. Furthermore, accumulation of Cytochrome c and COXIV immunoreactivity co-localized with TDP-43 aggregates in electroporated myofibers (Fig. 3d). Mitochondrial aggregates

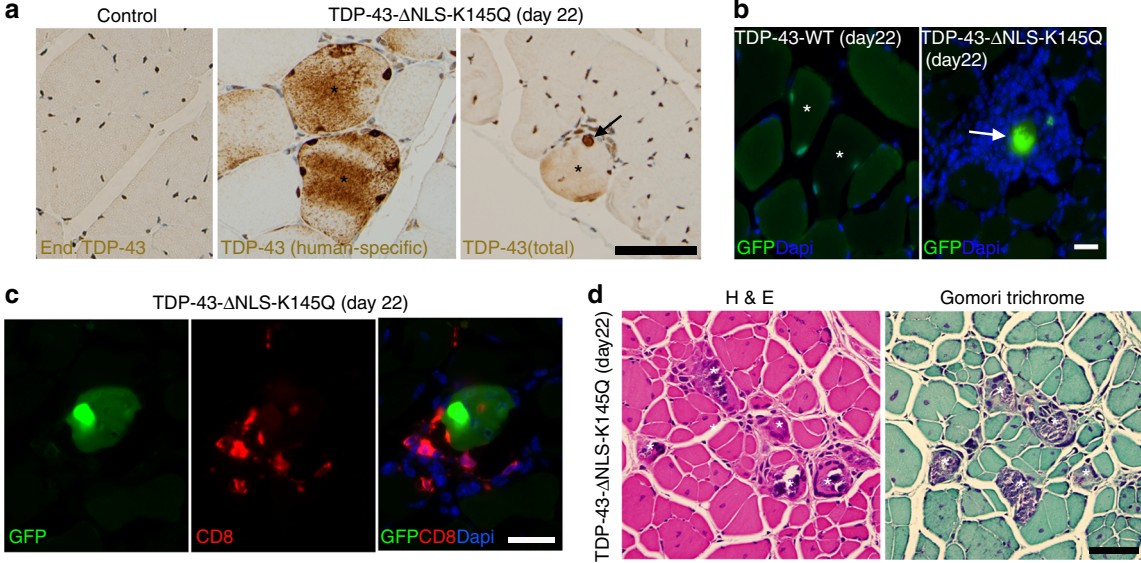

**Fig. 5** Chronic TDP-43 pathology that persists in skeletal muscles recapitulates aspects of sIBM pathology. Mouse TA muscles were mock treated (control) or electroporated with GFP-tagged TDP-43-WT or TDP-43-ΔNLS-K145Q for 22 days. **a** Muscle sections were analyzed by immunohistochemistry using a monoclonal TDP-43 antibody detecting the ectopically expressed TDP-43 construct (human-specific TDP-43 antibody) or a polyclonal TDP-43 antibody detecting total TDP-43 (endogenous mouse or ectopic human TDP-43). Both antibodies detected TDP-43 aggregates (*black asterisks* and *arrows*). Scale bar = 50 μm. Muscle sections were also analyzed by standard GFP fluorescence (**b**) or double-labeled with GFP (*green*) and CD8 (*red*) antibodies (**c**), while nuclei were stained with DAPI. Scale bar = 20 μm. **d** H&E and Gomori trichrome staining showed robust myofiber pathology associated with chronic TDP-43 aggregation (*white asterisks*). Scale bar = 50 μm

have been observed in several TDP-43 mouse models[49, 51], suggesting that abnormal TDP-43 conformation, whether weakly or strongly aggregated, may be sufficient to induce mitochondrial damage and activation of clearance mechanisms.

TDP-43 inclusions in sIBM, ALS, and FTLD-TDP tissues partly co-localize with other functionally linked ALS proteins that mediate proteasome or autophagic degradation[54, 55]. Cells (Fig. 4a) or myofibers (Fig. 4b) expressing aggregate-prone TDP-43-ΔNLS-K145Q were double-labeled and analyzed for recruitment of ubiquilin-2 (UBQLN-2) or optineurin (OPTN). Consistent with recruitment of degradative factors, cytoplasmic TDP-43 aggregates co-localized prominently with UBQLN-2 and OPTN (Fig. 4a, b, *white arrows*). We note that UBQLN-2 occasionally localized to smaller more discrete TDP-43 foci (Fig. 4a, see *white arrowheads*), potentially reflecting a pre-aggregated pool of TDP-43 targeted by UBQLN-2. The acetylation-mimic TDP-43 aggregates therefore generate overlapping co-pathologies that converge on several functionally related proteins implicated in ALS pathogenesis.

Since acetylation-mimic TDP-43 aggregates co-localized with p62, a sensitive and specific marker of sIBM[56], we further assessed whether ectopically generated muscle aggregates produced a sIBM myofiber phenotype. Normal endogenous TDP-43 is almost exclusively nuclear localized by immunohistochemistry (IHC) analysis, as expected (Fig. 5a). Prolonged expression of TDP-43-ΔNLS-K145Q for 22 days in vivo led to a striking accumulation of large TDP-43 inclusions within the myofiber sarcoplasm (Fig. 5a, *black asterisk*, and Fig. 5b, *white arrow*), suggesting an inability to clear mature TDP-43 aggregates. We observed accumulation of interstitial cells surrounding TDP-43-ΔNLS-K145Q electroporated myofibers, but not in TDP-43-WT expressing fibers (Fig. 5b). Double-labeling experiments suggested that infiltrating CD8+ T-cells were present surrounding aggregate-bearing myofibers (Fig. 5c), an inflammatory response that is implicated in myofiber damage in sIBM[12]. Finally, TDP-43-positive necrotic myofibers also

contained classic rimmed vacuoles that were observed by H&E and Gomori trichrome stains (Fig. 5d), indicating lysosomal dysfunction induced by the sustained accumulation of TDP-43 pathology.

To determine whether the aggregate burden compromised myofiber integrity, Evans blue dye (EBD) was intraperitoneal (i.p.) injected into mice prior to muscle tissue harvest. Upon membrane damage EBD is taken up into myofibers and readily detected by immunofluorescence. However, no EBD uptake was observed in TDP-43 transfected myofibers, and we observed very rare EBD-positive fibers in the entire muscle tissue irrespective of ectopic TDP-43 expression, consistent with a lack of overt muscle membrane damage induced by TDP-43 pathology (Supplementary Fig. 7, see *white arrows*). We conclude that acetylation-mimic TDP-43 recapitulates several key hallmarks of sIBM when introduced into muscle including TDP-43 aggregation, myofiber degeneration, and T-cell-mediated inflammatory responses.

**HSF1 suppresses acetylation-mimic TDP-43 pathology**. We next sought to identify clearance mechanisms as cytoprotective avenues to reduce acetylation-mimic TDP-43 pathology. HSF1, a master regulator of the cellular chaperone response, could facilitate the disaggregation of TDP-43 aggregates by virtue of its transcriptional induction of chaperone-dependent refolding. We took advantage of a previously engineered HSF1 mutant containing a single amino-acid substitution (L395E) and also lacking an auto-inhibitory domain (Δ221–315) that showed ~10-fold increased DNA-binding activity[36] (see Fig. 6a schematic). Wild-type HSF1, constitutively active HSF1 (Δ221–315, L395E), or an inactive DNA-binding-deficient HSF1 mutant (R71G) were individually co-expressed with TDP-43-ΔNLS-K145Q. Strikingly, wild-type HSF1 and more prominently active HSF1, but not the inactive HSF1, reduced insoluble phosphorylated TDP-43-ΔNLS-K145Q inclusions with no apparent effect on the soluble TDP-43 pool (Fig. 6b, c, Supplementary Fig. 8). These results were confirmed by immunofluorescence, which

showed ~90% reduction of phosphorylated TDP-43 inclusions in the presence of active HSF1 (Fig. 6d, e).

We next assessed whether delayed expression of HSF1 was sufficient to disaggregate pre-formed acetylation-mimic TDP-43 aggregates. Indeed, only introduction of the active HSF1 reduced the pre-formed insoluble TDP-43 pool (Fig. 6f, g, Supplementary

Fig. 8). Blocking either proteasome or autophagy-mediated degradation prevented HSF1-mediated disaggregation and restored insoluble TDP-43-ΔNLS-K145Q levels with no apparent effect on the soluble TDP-43 pool (Supplementary Fig. 9). Therefore, refolded insoluble TDP-43-ΔNLS-K145Q aggregates may be targeted for degradation via both pathways, with what

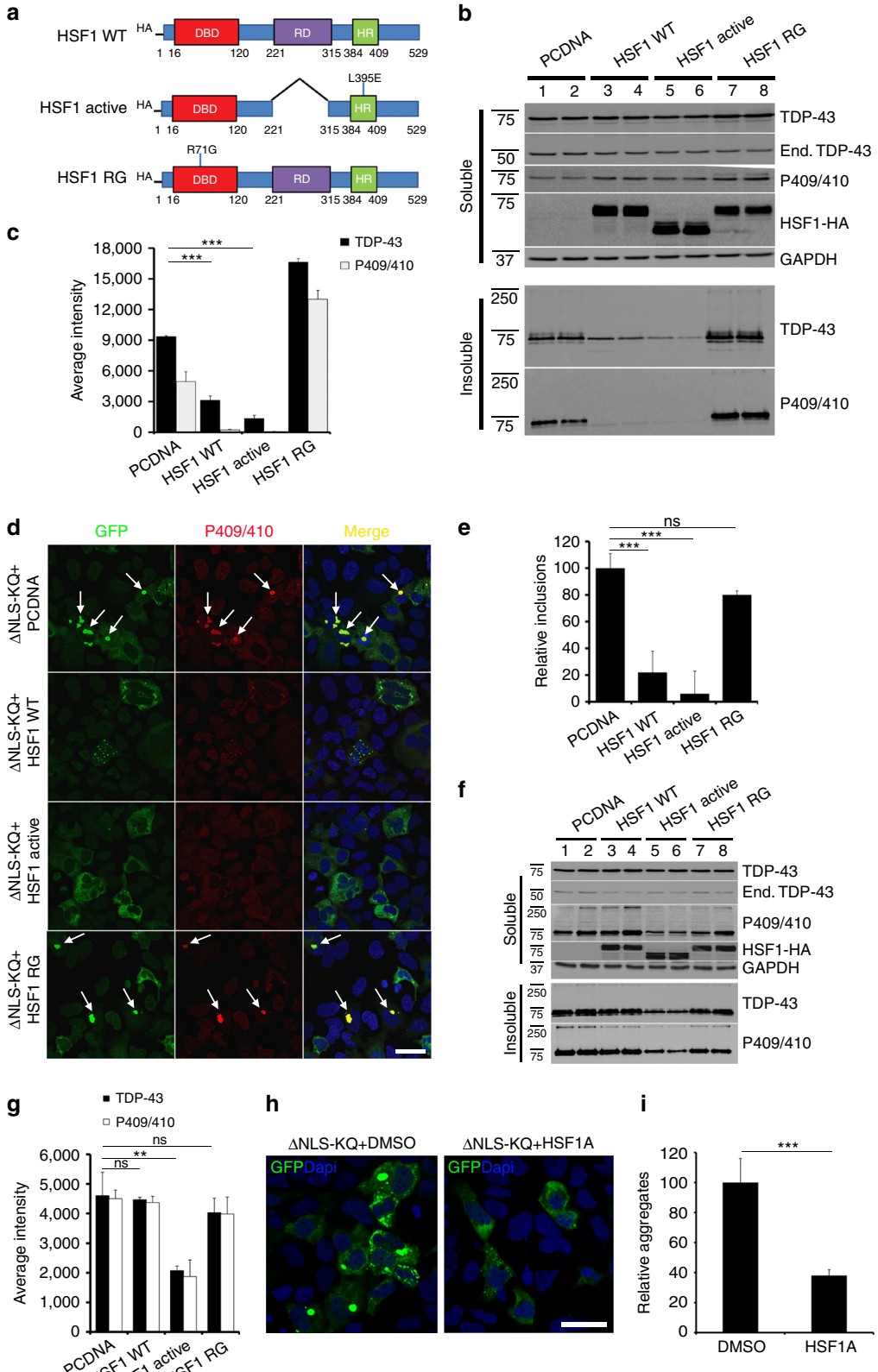

appears to be a more dominant role for the proteasome. To confirm these observations using a pharmacological approach, cells expressing acetylated TDP-43 aggregates were exposed to HSF1A, a compound that inhibits the TRiC protein complex thereby facilitating nuclear HSF1 accumulation, trimerization, and enhanced DNA binding[57]. HSF1A treatment for 24 h significantly reduced TDP-43 inclusion formation (Fig. 6h, i), suggesting that activation of endogenous HSF1 is sufficient to disaggregate TDP-43. Interestingly, suppression of TDP-43 aggregation was not observed in response to other reported HSF1 modulating compounds including the Hsp90 inhibitor 17-AAG and the current FDA-approved ALS drug riluzole (Supplementary Fig. 10)[58].

To assess whether HSF1 mediates TDP-43 aggregate clearance in vivo, we co-electroporated TDP-43-ΔNLS-K145Q with the indicated HSF1 constructs, or as a loss of function an HSF1-specific siRNA (Supplementary Fig. 11), into muscle fibers and TDP-43 aggregation was assessed by immunofluorescence. Similar to the cell-based approach, we observed significantly reduced overall TDP-43 aggregate burden in myofibers expressing active HSF1, and only small punctate TDP-43 foci remained (Fig. 7a, b). Importantly, these effects were not observed with the inactive HSF1 mutant. We further confirmed these observations using adjacently stained myofibers, in which HSF1 suppressed TDP-43 aggregation, but not in an adjacent myofiber that lacked HSF1 (Fig. 7c). Quantification of these results indicated that active HSF1 reduced TDP-43 aggregation in vivo by ~5-fold (Fig. 7d). While siRNA-mediated depletion of endogenous HSF1 in myofibers showed a trend toward increased TDP-43 aggregation, this was not statistically significant under the parameters and conditions tested (Fig. 7d).

**HSPs engage and suppress TDP-43 acetylation-mimic aggregates.** To investigate the targets of HSF1 that mediate TDP-43 disaggregation, we examined expression of a panel of HSPs that are transcriptionally induced by HSF1. We observed minimal induction of some HSPs (Hsp70, Hsp90, Hsp110), however, active HSF1 led to a more pronounced ~1.7–2.0 fold induction of Hsp27 and Hsp40 (Fig. 8a, b, Supplementary Fig. 12), two critical chaperones that are genetically linked to limb-girdle muscular dystrophy and primary motor neuropathy, respectively[59, 60]. While endogenous Hsp40, but not Hsp27, was physically recruited to cytoplasmic TDP-43 aggregates (Fig. 8c, see aggregates denoted by arrows), overexpression of either Hsp27 or Hsp40 was sufficient to suppress TDP-43 aggregation by immunofluorescence (Fig. 8d) and immunoblotting (Fig. 8e, Supplementary Fig. 12), which cleared the majority of TDP-43 aggregates, as only ~5–10% GFP-positive TDP-43 inclusions remained (Fig. 8f, g).

To assess whether Hsp27 or Hsp40 are necessary for HSF1-mediated suppression of TDP-43 aggregation, we examined whether siRNA-mediated knockdown of either chaperone could effectively prevent HSF1 from clearing TDP-43 aggregates (Fig. 9a, Supplementary Figs. 13 and 14). Hsp27 siRNA slightly restored insoluble TDP-43 levels (Fig. 9a, compare lane 1 to lanes 6–7), while Hsp40 siRNA alone, and to a greater extent the combined Hsp27/Hsp40 double siRNA, nearly completely restored TDP-43 aggregation and eliminated much of the disaggregation effects mediated by HSF1 (Fig. 9a, compare lane 1 to lanes 4–5 and 8–9). Quantification of these results suggests a synergistic involvement of both Hsp27 and Hsp40, with a more dominant role for Hsp40 in the clearance of TDP-43 aggregates (Fig. 9b).

Given the striking suppression of TDP-43 aggregation by Hsp40, we examined whether Hsp40 is linked to TDP-43 pathology in vivo by immunohistochemistry analysis of human ALS spinal cord. Hsp40 localization was diffusely cytoplasmic in the majority of ALS motor neurons with occasional focal, round, cytoplasmic immunoreactivity (Fig. 9c, *black arrows*). The Hsp40 immunoreactive structures were, in part, TDP-43-positive ALS lesions, as double-labeling revealed co-localization of Hsp40 with phosphorylated TDP-43 inclusions (Fig. 9c, see *white arrow* in merged image). Interestingly, in contrast to round TDP-43 inclusions, skein-like amorphous TDP-43 inclusions did not contain Hsp40 in the subset of ALS cases analyzed (Fig. 9c, see *white arrowhead*), suggesting that Hsp40 recruitment to TDP-43 aggregates may alter inclusion morphology and/or processing of its contents for degradation. Overall, our results support the notion that Hsp40 initially acts in a cytoprotective capacity to suppress toxic TDP-43 aggregation, but may be insufficient to counteract the full burden of TDP-43 pathology that accumulates in end-stage ALS spinal cord.

## Discussion

In this study, we provide strong evidence for acetylation-mimic induced full-length TDP-43 pathology in vitro and in vivo. A single acetylation-mimic (K145Q) that modulates TDP-43 RNA regulatory functions led to robust aggregation that was sufficient, when targeted to the cytoplasm, to trigger many of the pathological hallmarks associated with TDP-43 proteinopathy including hyper-phosphorylation, recruitment of functionally linked ALS proteins, perturbations of mitochondria, and an associated inflammatory signature. We provide evidence that enhanced chaperone function alleviated the burden of TDP-43 aggregates via an HSF-1 transcriptional cascade, suggesting an endogenous refolding program is capable of engaging and clearing TDP-43 inclusions in vivo. Our study suggests that pharmacological approaches to specifically target and enhance chaperone-dependent aggregate processing may restore proteostasis in individuals with ALS, sIBM, and other TDP-43 proteinopathies.

Our prior study suggested that acetylation achieved with the acetyltransferase CBP or K-to-Q mimics, but not K-to-R non-

**Fig. 6** An HSF1-dependent transcriptional program suppresses TDP-43 acetylation-mimic-induced aggregation. **a** Schematic diagram of HSF1 constructs highlighting active and inactive mutations. The regulatory domain (RD) was deleted and a single L395E was introduced to generate active HSF1, while R71G represents inactive HSF1. DBD DNA-binding domain, HR hydrophobic heptad repeat domain. **b** Cells were transfected with TDP-43-ΔNLS-K145Q in combination with the various HSF1 constructs (or control pCDNA vector) followed by biochemical fractionation and immunoblotting using the indicated TDP-43, P409/410, HSF1-HA, or GAPDH antibodies. Shown is a representative image containing duplicate samples from N = 3 independent experiments. **c** TDP-43 and phospho-TDP-43 levels from **b** were quantified by densitometry. **d** Cells expressing constructs similar to **c** above were analyzed by immunofluorescence using GFP and P409/410 antibodies. *White arrows* highlight phosphorylated TDP-43 aggregates. **e** Inclusion formation in **d** was quantified. **f** A sequential transfection was performed first with TDP-43-ΔNLS-K145Q followed by subsequent transfection of HSF1 for another 48 h. Soluble and insoluble fractions were analyzed by immunoblotting using the indicated antibodies. **g** Insoluble TDP-43 and phospho-TDP-43 were quantified, showing reduction of pre-formed TDP-43 aggregates by active HSF1. **h** Cells expressing TDP-43-ΔNLS-K145Q aggregates were exposed to DMSO or HSF1A and analyzed by immunofluorescence. **i** Aggregates were quantified and normalized to the number of total TDP-43 expressing cells. Error bars indicate SEM, and the *asterisk* indicates statistical significance with ***p-value < 0.001 and **p-value < 0.01, as measured by Student's t-test from N = 3 biological replicates. ns = not significant. Scale bar = 50 μm

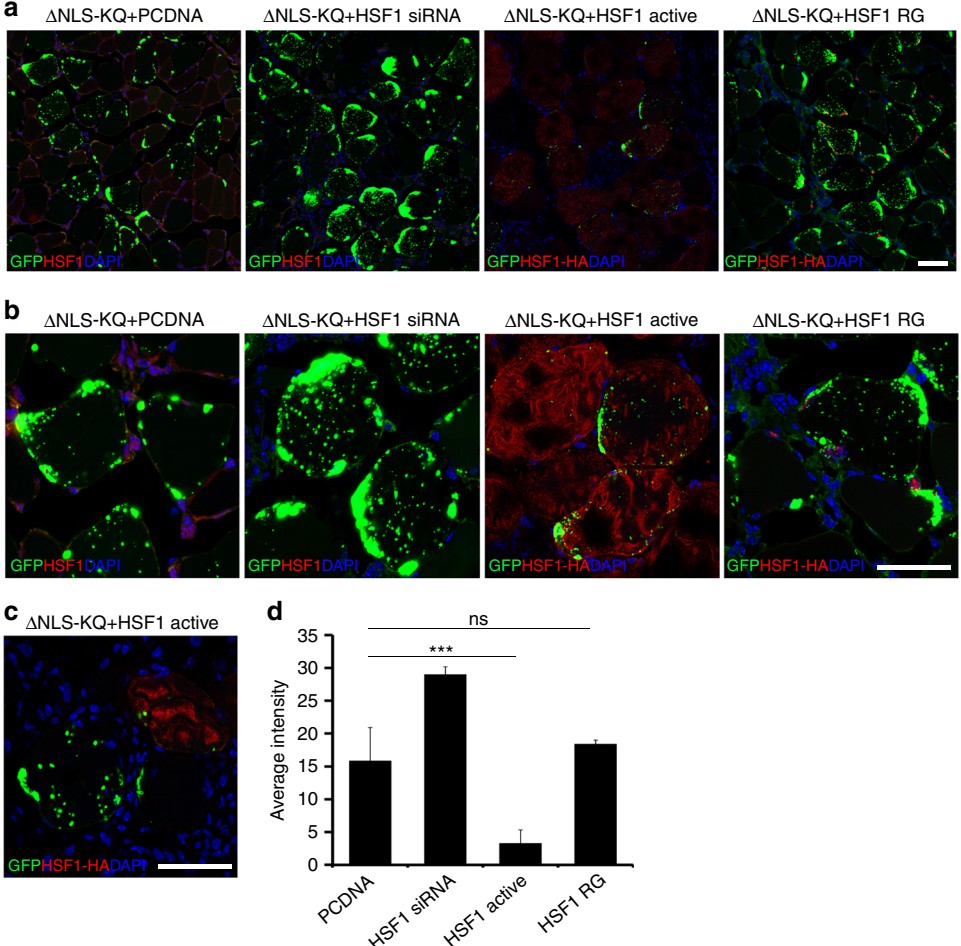

**Fig. 7** An HSF1-dependent cascade suppresses TDP-43 acetylation-mimic-induced pathology in vivo in skeletal muscle. **a** Immunofluorescence analysis was performed on TA muscles co-electroporated with GFP-tagged TDP-43-ΔNLS-K145Q in combination with either control vector (pCDNA), an HSF1-specific siRNA, active HSF1 or inactive HSF1 (R71G) for 4 days. **b** Higher magnification ×40 images of muscles described in **a** are shown. Reduced levels of aggregated TDP-43 were observed in the presence of active HSF1 (red). **c** Adjacent myofibers in the presence (red) or absence (white asterisk) of active HSF1 are depicted. Reduced levels of aggregated TDP-43 were observed in the myofiber expressing active HSF1. **d** TDP-43 average aggregation intensity was quantified among the total HSF1-expressing muscle fibers. Error bars represent SEM, and the triple asterisk indicates statistical significance with ***p-value < 0.001 as measured by Student's t-test from N = 3 biological replicates. ns = not significant. Scale bar = 50 μm

mimics, could abrogate TDP-43-RNA interactions leading to unstable TDP-43 conformation and aggregation[27]. However the significance of this observation in vivo remained unclear. Using electroporated skeletal muscle, acetylation-mimic TDP-43 generated a strong aggregate-prone phenotype, especially when targeted to the cytoplasm. This observation is consistent with more abundant TDP-43 pathology occurring in the cytoplasmic compartment. Recent studies in vitro have shown that loss of nucleic acid binding converts TDP-43 into unstable insoluble aggregates[61, 62], implying that physical association with nucleic acids may act to stabilize the TDP-43 C-terminal prion-like domain, leading to more soluble functional protein. Acetylation may therefore act as a regulatory switch to modulate TDP-43 affinity for target RNAs. Indeed, treatment of cells with agents that increase oxidative stress and promote loss of RNA binding led to acetylated and insoluble accumulation of TDP-43[27]. Residue K145 maps to an exposed lysine residue in close proximity to the DNA/RNA-binding interface[30], likely a more accessible region allowing rapid regulation of TDP-43 activity. Whether acetylation at other TDP-43 lysines also regulates TDP-43 aggregation is not currently known, but mass spectrometry analysis in vitro also detected K192 acetylation in RRM2[27].

Therefore, specific acetylation events within each RRM may converge to coordinately regulate TDP-43-binding affinity and hence its RNA regulatory functions.

While TDP-43 pathology correlates with degeneration in vivo, it is not entirely clear whether TDP-43 inclusions directly cause neurotoxicity. However, cytosolic TDP-43 accumulation likely drives an auto-regulatory cascade leading to further TDP-43 recruitment into cytosolic aggregates and loss of normal nuclear functions. TDP-43 inclusions therefore likely cause both loss and gain of function toxicity, however in vivo models of enhanced TDP-43 aggregation have been challenging to recapitulate. Recently, more robust full-length TDP-43 pathology in brain and spinal cord was achieved with NEFH promoter-driven expression of cytoplasmic TDP-43-ΔNLS leading to neurodegeneration and muscle denervation[63], indicating that higher level spinal cord expression may be required for more robust pathology. However, the authors noted that large inclusions typically seen in ALS patients were present in only a minority of motor neurons. Our ability to generate rapid, robust, and widespread TDP-43 pathology likely reflects a critical loss of nucleic acid binding associated with the K145Q acetylation-mimic mutation. The near absence of disease-causing mutations within the RRM domains

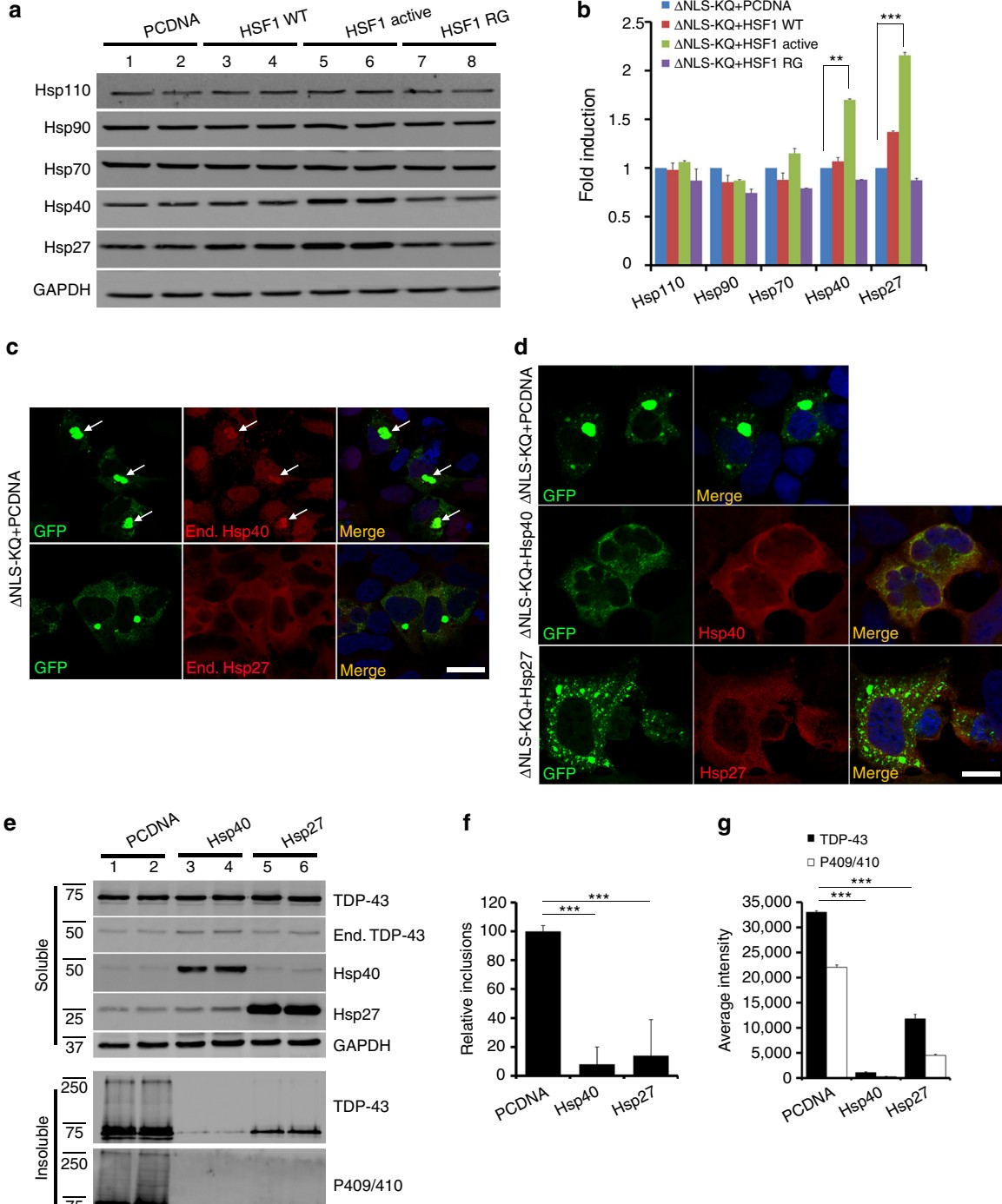

**Fig. 8** HSF1-regulated heat shock protein (HSP) expression suppresses TDP-43 acetylation-mimic-induced aggregation. **a** Cells transfected with TDP-43-ΔNLS-K145Q in combination with control vector (pCDNA), wild-type HSF1, active HSF1, or inactive HSF1 (R71G) were analyzed by immunoblotting using Hsp110, Hsp90, Hsp70, Hsp40, Hsp27, or GAPDH antibodies. **b** Fold induction of heat shock proteins from **a** above was quantified and normalized compared to the control. **c** Cells expressing TDP-43-ΔNLS-K145Q were analyzed by immunofluorescence to detect endogenous Hsp40 or Hsp27. We note that endogenous Hsp40, but not Hsp27, was recruited to TDP-43-ΔNLS-K145Q aggregates. *White arrows* highlight regions of co-localization. **d** Cells co-expressing TDP-43-ΔNLS-K145Q in combination with Hsp40 or Hsp27 constructs were analyzed by immunofluorescence. Reduced TDP-43 aggregation was observed in the presence of either Hsp40 or Hsp27. **e** Immunoblotting using the indicated TDP-43, P409/410, Hsp40, Hsp27, or GAPDH antibodies was performed on cells co-expressing TDP-43-ΔNLS-K145Q with control vector, Hsp40 or Hsp27. **f**, **g** Relative inclusion formation and levels of insoluble TDP-43 in the presence or absence of Hsp40 or Hsp27 was determined. Error bars indicate SEM, and the *asterisks* indicate statistical significance with ***p-value < 0.001 and **p-value < 0.01, as measured by Student's t-test from N = 3 biological replicates. ns = not significant. Scale bar = 50 μm

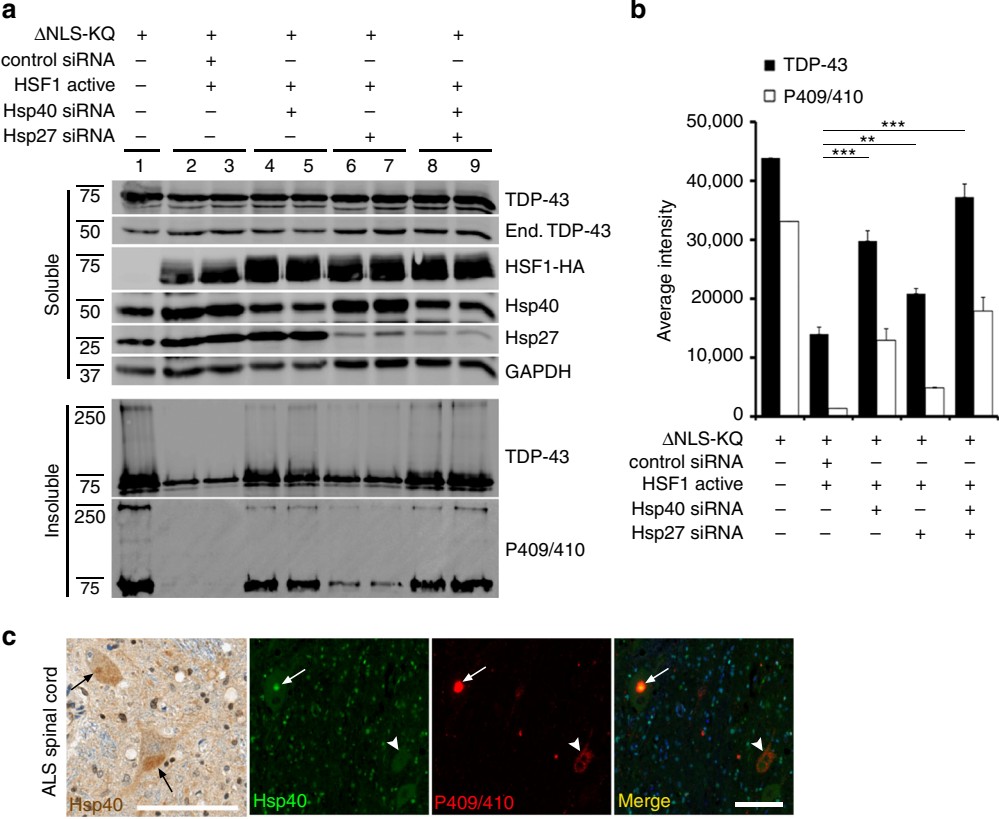

**Fig. 9** Depletion of Hsp27 and Hsp40 impairs HSF1-mediated disaggregation of acetylation-mimic TDP-43. **a** Cells co-transfected with TDP-43-ΔNLS-K145Q (ΔNLS-KQ) and active HSF1 were treated with control siRNAs or specific siRNAs targeting either Hsp40 or Hsp27 and subsequently analyzed by immunoblotting using TDP-43, P409/410, HSF1, Hsp40, Hsp27, and GAPDH antibodies. **b** The average aggregate intensity was quantified to assess the insoluble TDP-43 levels that were restored by Hsp27 or Hsp40 siRNA treatments. Error bars indicate SEM, and the *asterisk* indicates statistical significance with ***$p$-value < 0.001 and **$p$-value < 0.01, as measured by Student's $t$-test from $N = 3$ biological replicates. ns = not significant. **c** Lumbar ALS spinal cord was analyzed by immunohistochemistry using an Hsp40 antibody or double-labeled using Hsp40 and phosphorylated TDP-43 (P409/410) antibodies. Nuclei were stained with DAPI. The *black arrows* indicate focal, cytoplasmic, round Hsp40 accumulation. The *white arrows* highlight co-localization of Hsp40 with phosphorylated TDP-43 aggregates, while the *white arrowhead* indicates a skein-like amorphous TDP-43 inclusion that is devoid of Hsp40. Scale bar = 100 μm

(or surrounding the K145 residue) could reflect lethality associated with sustained RNA-binding deficiency, which conceivably would impact the splicing and/or expression of up to ~6000 TDP-43 targets in neurons, many of which are essential for synaptic function[19]. In this regard, there is debate about whether TDP-43 binding to downstream targets promotes or alleviates toxicity, as several studies indicate that loss of RNA binding can suppress TDP-43-mediated toxicity[64, 65]. In contrast, our results support the notion that acetylation induces loss of RNA binding, enhanced aggregation, and progressive degeneration (Fig. 5). However, we cannot currently exclude the possibility that acetylated TDP-43, at least in part, acquires new RNA-binding properties, as previously proposed for some ALS-associated TDP-43 mutations[66].

HSF1 has been implicated in aggregate clearance and neuroprotection, particularly in SOD1-dependent ALS models leading to delayed disease onset and progression[37, 67]. In addition, HSF1 activating compounds including arimoclomol reduced SOD1 aggregation and were neuroprotective[68]. Our results suggest that TDP-43 is also targeted by an endogenous refolding program, which is consistent with two recent studies[43, 44]. However, HSF1 may be kept partially inactive in tissues that are uniquely vulnerable to TDP-43 pathology including skeletal muscle and CNS tissues. Supporting this possibility, neurons were shown to possess an unusually high threshold for HSF1 activation and

subsequent HSP induction[40]. Thus, enhanced pharmacological HSF1 activation may have therapeutic value in ameliorating TDP-43 pathology. Since sustained HSF1 activity is associated with tumor metastasis and progression[69], an alternative approach could target critical downstream chaperones including small HSPs (Hsp27) and Hsp40 that are most highly induced by HSF1 and capable of suppressing TDP-43 aggregates (Fig. 8)[43, 44]. While the roles of these HSPs in TDP-43 proteinopathies are not fully understood, genetic mutations in DNAJB1 and HSPB1 that encode Hsp40 and Hsp27 are associated with limb-girdle muscular dystrophy and motor neuropathies[59, 60]. Furthermore, Hsp27 expression alone is sufficient to delay motor phenotypes and increase motor neuron survival in SOD1-G93A mice[70]. Thus, proper Hsp27/40 expression in combination with proper subcellular localization and target engagement (i.e., TDP-43 aggregates) may be critical to achieve optimal aggregate suppression. Future efforts to modulate HSPs in the recently described TDP-43 transgenic mice[63] may provide insight into HSP function in the context of more robust brain and spinal cord pathology.

While HSF-1 led to disaggregation of pre-formed TDP-43 aggregates, residual TDP-43 aggregates still remained (Fig. 6f, g), indicating that synergistic approaches to boost chaperone activity may be advantageous. For example, recent efforts to engineer variants of the AAA+ ATPase Hsp104 have proven extremely

effective against a wide range of protein aggregates including tau, α-synuclein, and TDP-43[71]. While our data suggest that HSPs mediate most of the protective effects of HSF1, non-HSPs downstream of HSF-1 may also be relevant in mediating TDP-43 disaggregation[38]. In this regard, transcriptomic approaches may be necessary to clarify the range of specific HSF1 targets that are inducible in brain, spinal cord, or skeletal muscle in vivo. We speculate that vulnerability to TDP-43 aggregate formation may be explained by a lack of sufficient HSF1 levels and/or function in vivo. Therefore, efforts to augment or restore HSF1 activity or downstream chaperone function could provide therapeutic opportunities to combat a range of multisystem TDP-43 proteinopathies affecting both CNS and non-CNS tissues alike.

## Methods

**Plasmids and cell culture.** QBI-293 cells (MP Biomedicals) and Neuro2A cells (ATCC) are commercially available. NSC-34 cells were originally purchased from Tebu-Bio and kindly provided by Dr. Nikolay Dokholyan (University of North Carolina). All mammalian expression plasmids were amplified in DH5α bacteria (New England Biolabs). TDP-43 constructs were generated using either pEGFP-C1 or DsRed-C1 vectors (Clontech) and K-to-Q mutations at residue K145 were created by site-directed mutagenesis (Quikchange kit; Stratagene) using primers listed in Supplementary Table 1. Human HSF1 (hHSF1) constructs were kindly provided by Dr. Dennis Thiele (Duke University) and mutated to active and inactive forms using the same strategy. The regulatory domain (RD) was deleted and a single L-to-E mutation at residue L395 was introduced to generate the active form of HSF1, while R71G was generated for inactive HSF1. Human optineurin (OPTN) and ubiquilin-2 (UBQLN-2) expression plasmids were purchased from Addgene. Plasmids were transfected into QBI-293 or NSC-34 cells using Fugene 6 (Promega) or Lipofectamine 2000 (Thermo Fisher Scientific) following the manufacturer's instructions. For HSP siRNA treatment, cells were cultured in six-well plates for 24 h prior to transfection. Co-transfection of multiple plasmids (e.g., TDP-43-ΔNLS-K145Q combined with control PCDNA or HSF1-active) or siRNA duplexes (targeting Hsp40 or Hsp27, Santa Cruz Biotechnology) was performed using Lipofectamine 2000 per the manufacturer's protocols. All siRNAs were transfected twice sequentially to achieve optimal siRNA knockdown (50 pmol incubated for 24 h performed twice).

For HSF1 siRNA treatment (Supplementary Table 1), Neuro2A cells were treated with control or a panel of HSF1 siRNA for 72 h. Cells were harvested and analyzed by immunoblotting using an HSF1 antibody (Enzo Life Sciences). For sequential transfection, QBI-293 cells were first transfected using Fugene 6 with TDP-43-ΔNLS-K145Q for 18 h, and then transfected again with control vector, wild-type HSF1, HSF1 active, or HSF1 R71G mutant plasmids for another 48 h. Cells were harvested and analyzed by immunoblotting using the indicated antibodies as described in text. For HSF1A (Axon Medchem) drug treatment, QBI-293 cells were grown on poly-D-lysine-coated coverslips and transfected as described above, treated with 10 μM HSF1A or DMSO for 24 h and analyzed by confocal microscopy. For riluzole and 17-AAG treatments, cells were treated with riluzole (5 μM) or 17-AAG (5 μM) (Sigma) for 24 h and analyzed by immunoblotting. For MG-132 or 3-methyladenine (3-MA) drug treatments, QBI-293 cells were cultured in six-well plates for 24 h prior to transfection. TDP-43-ΔNLS-K145Q was transfected first, then active HSF1 was added ~12 h after the initial transfection. After 48 h, cells were treated with MG-132 (1 μM) or 3-MA (10 mM) for 24 h. Cells were harvested and analyzed by immunoblotting using the indicated antibodies, as described in the text. For NSC-34 cell culture experiments, cells were maintained in high-glucose DMEM formulation (D5796, Sigma) supplemented with 10% FBS. NSC-34 transfections were performed on poly-D-lysine-coated coverslips, and plasmids were transfected for 48 h using Lipofectamine 2000 (Life Technologies) following the manufacturer's protocols, followed by confocal microscopy, as described below.

**Skeletal muscle electroporation.** All experiments were performed in strict compliance with animal protocols approved by the Institutional Animal Care and Use Committees (IACUC) of the University of North Carolina at Chapel Hill (under approved protocol 14.107.0). C57BL6 mice were obtained from Charles River Laboratories and used for this study. Mouse muscle electroporations were performed based on a high-efficiency protocol that was developed in skeletal muscle[46]. The detailed procedure is as follows: Wild-type 12-week-old C57BL/6 mice (male and female) were treated with hyaluronidase, 5 mg ml$^{-1}$ in 0.9% saline 1 h before injection. TA muscles were then injected with 50 μl hyaluronidase per leg and treated for 45 min. The mice were anesthetized using Avertin (1.25% in phosphate-buffered saline (PBS), Sigma) and fur was removed with Nair. Using 25 μl microsyringe (VWR), plasmids (25 μl of 0.2–0.4 μg μl$^{-1}$) were injected into TA muscles. A BTX 830 (BTX Harvard Apparatus) machine was set to 50 V, 5 pulses, pulse length 60 ms, interval time 200 ms. Electrodes (tweezertrodes model 520) were applied to muscle belly and then again in a perpendicularly manner. Mice were allowed to fully recover overnight. All electroporated animals were standard

wild-type mice on the same background, no randomization was used in these animal experiments, and staining analysis was performed in a blinded fashion. A minimum of N = 3–5 mice (hind-limb leg electroporation of TA muscles) were used for each individual electroporation experiment, which was sufficient to provide adequate power for these experiments (0.80). In the event that protein expression was not optimal within our detection limits (using biochemical or histological methods) due to technical issues or limitations, samples were excluded from our analysis.

**Biochemical analysis of cells and muscle tissue.** Biochemical analyses for preparation of lysates were performed as follows: Cells from six-well culture dishes or mouse TA muscles were sonicated and homogenized in RIPA buffer (50 mM Tris pH 8.0, 150 mM NaCl, 1% NP-40, 5 mM EDTA, 0.5% sodium deoxycholate, 0.1% SDS) supplemented with 1 mM phenylmethylsulfonyl fluoride (PMSF), a mixture of protease inhibitors (1 mg ml$^{-1}$ pepstatin, leuptin, N-p-tosyl-L-phenylalanine chloromethyl ketone, Nα-Tosyl-L-lysine chloromethyl ketone hydrochloride, trypsin inhibitor; Sigma) and a mixture of phosphatase inhibitors (2 mM imidazole, 1 mM NaF, 1 mM sodium orthovanadate; Sigma). Lysates were then sonicated and centrifuged at 21,100×g for 30 min at 4 °C, and then rinsed in RIPA buffer to deplete the soluble protein pool. The insoluble pelleted fractions were then extracted in 75 μl (cells) or 250 μl (muscle tissues) urea buffer (7 M urea, 2 M Thiourea, 4% CHAPS, 30 mM Tris, pH 8.5), sonicated, and centrifuged at 21,100×g for 30 min at room temperature. All soluble and insoluble fractions were subsequently analyzed by western blotting using the following antibodies: rabbit polyclonal anti-TDP-43 (Proteintech), rabbit polyclonal anti-phospho-TDP-43 (409/410) (Proteintech), rabbit polyclonal anti-GAPDH (Santa Cruz Biotechnology), mouse mono, polyubiquitinylated conjugates (FK2, Enzo Life Sciences), purified mouse anti-P62 (BD Transduction Laboratories), rabbit monoclonal anti-COXIV (Cell Signaling), rabbit polyclonal anti-TOM20 (FL-145, Santa Cruz Biotechnology), mouse anti-Cytochrome c (BD Pharmingen), rabbit polyclonal anti-HA (Y-11, Santa Cruz Biotechnology), rabbit polyclonal anti-Hsp110 (Enzo Life Sciences), rat monoclonal anti-Hsp90 (16F1, Enzo Life Sciences), mouse monoclonal anti-Hsp70/Hsp72 (C92F3A-5, Enzo Life Sciences), rabbit polyclonal anti-Hsp40/Hdj1 (Enzo Life Sciences), goat polyclonal anti-Hsp27 (M-20, Santa Cruz Biotechnology), rat monoclonal anti-HSF1 (10H8, Enzo Life Sciences). Dilution ratios were 1:1000 for all the antibodies unless otherwise stated. Western blot quantification was measured by densitometry and analyzed using ImageQuant TL 1D v8.1 (GE Healthcare Life Sciences).

**Immunocytochemistry and quantification.** QBI-293 cells were grown on PDL-coated coverslips and transfected as described above and cultured for the indicated times. The cells were irreversibly fixed in 4% paraformaldehyde (PFA) for 10 min, rinsed in PBS (three times) and permeabilized once with 0.2% Triton X-100 (Sigma) in PBS for 10 min. Fixed cells were then blocked in 2% milk for 1 h and incubated with specified primary antibodies overnight at 4 °C. Cells were washed in PBS and incubated with Alexa-488 or 594-conjugated secondary antibody. Nuclei were counterstained with DAPI (Sigma). Cells were analyzed using a LSM780 confocal laser microscope (Carl Zeiss). The relative inclusions counts were normalized to that of cells co-transfected with control vector (PCDNA 3.1) or DMSO mock treatment (in the case of HSF1A exposure), using >5 fields and a minimum of 200 transfected cells. Sampling error was determined using the SEM. Statistical analysis was performed with a two-tailed unpaired t-test with unequal variance (significance set at p-value < 0.05). All of the quantitative immunofluorescence analyses were independently validated and confirmed with at least N = 3 biological replicates. Primary antibodies used for immunofluorescence were as follows: rabbit polyclonal anti-GFP (FL, Santa Cruz Biotechnology), mouse monoclonal anti-GFP (B2, Santa Cruz Biotechnology), rabbit polyclonal anti-phospho-TDP-43 (409/410) (Proteintech), mouse mono, polyubiquitinylated conjugates (FK2, Enzo Life Sciences), rabbit polyclonal anti-HA (Y-11, Santa Cruz Biotechnology), rabbit polyclonal anti-proteasome 20 S core subunit (Enzo Life Sciences), mouse monoclonal anti-SQSTM1(Abnova), mouse monoclonal anti-UBQLN2 (555–624, Abnova), rabbit polyclonal anti-Hsp40/Hdj1 (Enzo Life Sciences), goat polyclonal anti-Hsp27 (M-20, Santa Cruz Biotechnology). Antibody dilutions were 1:500 for all antibodies unless otherwise noted.

**Histological analysis of skeletal muscle.** Electroporated TA muscles were harvested and immediately immersed in 4% PFA overnight at 4 °C, and washed with 1×PBS to remove traces of paraformaldehyde. The muscle tissues were then transferred to 15% sucrose for OCT immersion embedding and sectioned into 10 μm serial sections. Double-labeling immunofluorescence (IF) was performed using standard protocols. Briefly, sections were defrosted for 5 min, and fixed in 4% PFA for 10 min. Sections were then washed with PBS three times for 5 min each, and incubated in PBST (PBS and 0.1% Triton) for 15 min. The sections were further blocked with PBTN (10% goat serum in PBST) for 1 h at room temperature in humidified chamber, and subsequently incubated with primary antibodies of interest overnight at 4 °C. Sections were washed and incubated with Alexa-488- or 594-conjugated secondary antibodies (Molecular Probes). Nuclei were counterstained with DAPI (Sigma). After washing in PBS, sections were coverslipped with Fluoromount-G mounting medium (SouthernBiotech).

CD8+ cytotoxic T-cell staining was performed by the Animal Histopathology Core at the University of North Carolina. Sections were placed in acetone for 2 min and then dried at room temp for 5–10 min. After 10 min PBS wash, the sections were transferred to 10% formalin for 5 min and rinsed with PBS for 10 min. A hydrogen peroxidase step was performed for 8 min and rinsed with PBS for 10 min. The sections were then incubated with the primary CD8 antibody for 2 h at room temperature and washed with PBS. Secondary anti-rat antibody (Omnimap) was applied for 32 min at room temperature and rinsed. The sections were further rhodamine stained for 16 min and rinsed. The sections were stained with Hoescht 33258 (Invitrogen solution) for 7 min and coverslipped using Prolong Gold Antifade reagent (P36934, Life Technologies).

Images were obtained in a blinded manner (electroporated plasmids were unknown at the time of imaging) using either an Olympus XI83 microscope or a LSM780 confocal laser microscope (Carl Zeiss). The following primary antibodies were used for IF analysis: rabbit polyclonal anti-GFP (FL, Santa Cruz Biotechnology), mouse monoclonal anti-GFP (B2, Santa Cruz Biotechnology), rabbit polyclonal anti-phospho(409/410)-TDP-43 (Proteintech), mouse mono, polyubiquitinylated conjugates (FK2, Enzo Life Sciences), rabbit polyclonal anti-HA (Y-11, Santa Cruz Biotechnology), rabbit polyclonal anti-proteasome 20S core subunit (Enzo Life Sciences), mouse monoclonal anti-SQSTM1(Abnova), mouse monoclonal anti-UBQLN2 (555–624, Abnova), rat monoclonal anti-HSF1(10H8, Enzo Life Sciences), rabbit polyclonal anti-Hsp40/Hdj1 (Enzo Life Sciences), goat polyclonal anti-Hsp27 (M-20, Santa Cruz Biotechnology), rat anti-mouse CD8 (eBioscience, 1:100). The average signal intensity of muscle sections was quantified using Image J software. Three fields were examined, and sampling error was calculated using the SEM. Statistical analysis was determined using a two-tailed unpaired $t$-test with unequal variance (significance set at $p$-values <0.05). All quantitative fluorescence was independently validated with a minimum of $N = 3$ biological replicates.

Immunohistochemical muscle staining was performed on paraffin-embedded TA muscle tissues. Antigen retrieval was performed with a citrate based buffer (Discovery CC2, 760–107, pH 6.0) for 64 min at 100 °C and then blocked with hydrogen peroxidase for 12 min at room temperature. The slides were incubated for 1 h at room temperature in a mixture of the primary antibody, a tris-based diluent (Discovery PSS Diluent, 760–212), and the DAKO ARK's Biotinylated and Blocking Reagents (Dako ARK, Animal Research Kit), followed by Streptavidin-HRP incubation (DAKO ARK) for 32 min at room temperature. The slides were treated with 3,3′-diaminobenzidine (DAB) for visualization and Hematoxylin counterstain for 12 min, and then Bluing Reagent for 4 min. The staining was performed using Ventana's Discovery Ultra Automated IHC staining system. Primary antibodies used for IHC analysis were as follows: mouse monoclonal anti-TDP-43 (human-specific TDP-43 antibody) (Proteintech, 1:250), rabbit polyclonal anti-TDP-43 (total mouse and human TDP-43 antibody) (Proteintech, 1:500). Images were obtained using an Olympus BX41 light microscope.

**Injection of Evans blue dye**. Mouse TA muscles were electroporated with TDP-43-ΔNLS-K145Q plasmid for a total of 22 days. One day prior to muscle harvest, the mice were i.p. injected with 1% EBD (Sigma) (w/v) in PBS(pH 7.5). After EBD injection, animals were returned to their cage and allowed to recover overnight. TA muscles were harvested after 24 h of EBD injection and analyzed by immunostaining using the indicated antibodies described in text.

**Hematoxylin and eosin and Gomori trichrome staining**. Mouse TA muscles electroporated for the indicated times were harvested and immediately immersed in 4% PFA overnight at 4 °C, washed with 1×PBS to remove traces of PFA and paraffin-embedded. H&E staining was performed using standard protocol for paraffin-embedded muscle tissue sections at 4 μm thickness. Briefly, unstained slides were placed in an oven heated to 65–73 °C for 20–25 min to remove excess paraffin. Paraffin was removed using xylene followed by ethanol dehydration. The sections were rehydrated with tap water and stained with hematoxylin (Richard-Allan Scientific) for 2 min. Richard Allan Scientific Clarifier 2 was applied for 1 min. After three times water wash, Bluing Reagent (Richard-Allan Scientific) was applied for 30 s and washed with water for 2.5 min. After 95% ethanol dehydration, the eosin–phloxine stain (Richard-Allan Scientific) was applied for 1 min. The sections were then dehydrated in ethyl alcohol, cleared with xylene, and coverslipped using a Leica Surgipath micromount medium (Leica Surgipath). For Gomori trichrome staining, coverslips with sections were placed in a ceramic staining rack (Thomas Scientific) and immersed in hematoxylin for 5 min. Sections were immersed in Gomori trichrome for 10 min and differentiated using 0.2% acetic acid. The sections were then dehydrated in ethyl alcohol, cleared with xylene, and coverslipped using a Leica Surgipath micromount medium. Images were obtained using an Olympus BX41 light microscope.

**Histological analysis of ALS spinal cord**. ALS spinal cord sections were kindly provided by Dr John Q. Trojanowski (University of Pennsylvania) from two patients harboring abundant acetylated TDP-43 pathology[27].

Hsp40 immunohistochemical analysis and dual-color IF staining of Hsp40 (rabbit polyclonal anti-Hsp40/Hdj 1, Enzo Life Sciences, 1:150) and phospho-TDP-43 was performed by the UNC Translational Pathology Laboratory. Slides were dewaxed in Bond Dewax solution (Leica Biosystems) and hydrated in Bond Wash

solution (Leica Biosystems). Antigen retrieval for both stains were done for 30 min in Bond-epitope retrieval solution 1 (pH 6.0, Leica Biosystems), followed with 5 min endogenous peroxidase blocking with Bond peroxide blocking solution (Leica Biosystems). The protein blocking reagent (Leica Biosystems) was added for 10 min. Primary antibodies were applied for 1 h. Chromogenic detection of Hsp40 was done using Bond Polymer Refine Detection system with DAB visualization and Hematoxylin counterstain. Stained slides were dehydrated and coverslipped. For dual-color IF staining, first phospho-TDP-43 was detected with rabbit anti-rat IgG (Vector labs), Bond polymer (Leica Biosystems) and Tyramide -Cy5 (Perkin Elmer), then with Hsp40 with Bond polymer and Tyramide Cy3 (Perkin Elmer). After completion of the first stain and before addition of the second primary antibody the appropriate antigen retrieval protocol and peroxide blocking steps as described above were applied. Slides were counterstained with Hoechst (Invitrogen) and mounted with ProLong Gold antifade reagent (Life Technologies). IHC stained slides were digitally imaged in Aperio ScanScope XT (Leica Biosystems) using ×20 objective. High-resolution acquisition of double-color IF slides in the DAPI, Cy3, and Cy5 channels was performed in Aperio ScanScope FL (Leica Biosystems) using ×20 objective.

**Immunoprecipitation and mass spectrometry analysis**. Immunoprecipitation followed by mass spectrometry was performed as follows. For cell-based mass spectrometry analysis, 10 cm dishes of QBI-293 cells were transiently transfected with TDP-43-ΔNLS (acetylation mapping) or TDP-43-ΔNLS-K145Q (ubiquitination mapping) plasmids. Cell lysates were generated and total TDP-43 was immunoprecipitated using anti-TDP-43 antibody (clone 171), which was complexed to protein A/G beads (Sigma). The immunoprecipitated TDP-43 was analyzed either directly without SDS-PAGE, or alternatively analyzed by SDS-PAGE electrophoresis and Coomassie staining, and finally gel excised for analysis by nanoLC/nanospray/MS/MS. Acetylated lysine mapping was performed at the University of Pennsylvania proteomics core facility using Xcalibur 2.0 (ThermoFisher), PEAKS 6.0 (Bioinformatics Solutions Inc.), and Scaffold 3 (Proteome Software) software packages. The significance cutoffs used were as follows: peptide $p$-value < 95%; protein $p$-value < 99.0%. TDP-43 ubiquitination site mapping was performed with both discovery and target proteomics using a quadrupole orbitrap mass spectrometer operating in a traditional data-dependent top 12 mode and in parallel reaction monitoring mode (PRM), respectively. PRM data were analyzed in Skyline.

**Data availability**. All relevant data are available from the corresponding author upon reasonable request.

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

## Acknowledgements

Support for this work was provided by the NIH grants R00-NS080985 (T.J.C.) and P30-ES025128 (M.S.B). We thank Dr. John Q. Trojanowski (University of Pennsylvania) for providing ALS spinal cord tissue as well as the patients and their families for making the research here possible. We thank Dr. Dennis Thiele (Duke University) for providing human HSF1 plasmids. We thank Dr. William Snider and Dr. Vladimir Ghukasyan (University of North Carolina, Chapel Hill) for helpful discussions and assisting with tissue imaging and processing. We also thank Dr. C. Ryan Miller, the UNC Translational Pathology Laboratory (TPL), and the UNC Animal Histopathology Core for their technical support and very insightful suggestions and troubleshooting tips throughout these studies.

## Author contributions

P.W. performed the majority of the experiments, data analysis, and helped in writing the manuscript. C.M.W. performed the cell-based experiments and subsequent immuno-blotting. C.-X.Y. and M.S.B. conducted mass spectrometry analysis and interpreted data to identify and quantify acetylated and ubiquitinated TDP-43 residues. T.J.C. directed the study and wrote the manuscript. All authors contributed to the final version of the manuscript.

## Additional information

**Competing interests:** The authors declare no competing financial interests.

