## [Peer Review file · Nature Communications]

Reviewers' comments:

Reviewer #1 (Remarks to the Author):

This group has previously found that TDP-43 can be acetylated at specific lysine residues. The present follow-up work initially confirms acetylation at K145 in the absence of ectopic acetyltransferase, a desired control experiment. At least for transfected TDP-43 protein with mutated nuclear localization sequence (Δ NLS) in QBI-293 cells. The authors go on with GFP fusion proteins to show in transfected QBI-293 cells an in electroporated muscle that supposedly acetylation mimicking K145Q substitutions enhance TDP-43 aggregation, particularly when combined with Δ NLS. Co-localization of the resulting inclusions with several components of the ubiquitin-proteasome system is taken as evidence for impaired protein clearance. Finally, hypothesis-driven examination of an involvement of HSF1 is tested, and systematic analysis of HSF1-dependent chaperones reveals the participation of 40kDa and 27kDa heat shock proteins.

The manuscript is well written and clear to follow, including the methods descriptions. Overall the results support the conclusions and the interpretations are reasonable. Although a follow-up on previously reported TDP-43 acetylation and aggregation, the present study provides novel detail information for the cognate chaperone defense program. The in vivo part is particularly relevant for the muscle TDPopathy, IBM.

Specific Comments:

1. More stringent language would use the term "acetylation-mimic" instead of "acetylated" when describing properties of the K->Q substituted TDP-43. Please check the manuscript text throughout.
2. The staining patterns of TDP-43 GFP fusion proteins in muscle should be described more carefully. First of all, endogenous TDP-43 staining in control tissue must be shown as a reference. Why does KQ-TDP-43-WT appear nuclear excluded, in stark contrast to the staining pattern in 293 cells? Why does the Δ NLS mutant show a staining pattern near nuclei and not distribute further out into the cytosol?
3. The quality of several dual-label immunostainings is insufficient.
 - a. Fig. 2c phospho-TDP-43 immunostaining suffers from high background. Much "red" is in areas not seen by GFP epifluorescence.
 - b. Similar criticism for the ubiquitin co-stain, although here higher "background" signals may be expected. Nevertheless, several ubiquitin "inclusions" are not mirrored in the green channel (Fig. 3a).
 - c. In figure 4a there must be channel bleeding. Or why would ubiquitin and optineurin become detectable only in transfected cells?
 - d. Some longitudinal sections may be illustrative to see the nucleocytoplasmic distributions along the myofiber.
4. It is not quite clear what to make of the Western blot smears of mitochondrial marker proteins. Please show immunostainings. And refer to precedent literature, if applicable.
5. Please show controls in Fig. 5a,b. What is intended to demonstrate with Fig. 5d, right panel?
6. Supplementary Figure 5 and 6 are swapped.
7. Page 11, line 21: "...depletion of endogenous HSF1 did not significantly alter TDP-43 aggregation..." why so cautious here? One does get a visual impression of more aggregation (?)

8. Discussion: Acetylation of K145 within RRM1 could affect RNA binding, etc. Of course, this reasonable assumption would be better supported by experimental evidence.

Reviewer #2 (Remarks to the Author):

In this interesting study, Cohen and colleagues provide evidence consistent with acetylation-induced TDP-43 aggregation in cells and in vivo. This aggregation can be buffered and even partially reversed by introduction of activated HSF1, which elicits a transcriptional program of heat-shock proteins. Of these, Hsp27 and Hsp40 appear to be particularly important for antagonizing aggregation. Interestingly, it is possible to elicit these same effects pharmacologically using HSF1A a compound developed by the Thiele lab to induce a heat-shock response by engaging TriC. Overall, this study is interesting and deserves to be published, however a question remains that should be addressed prior to publication:

1. Evidence is presented suggesting that TDP-43 inclusions can be disaggregated by HSF1-induced chaperones. However, what is the fate of the disaggregated TDP-43? Is it refolded or does it get degraded? This question could be addressed by shutting down TDP-43 expression (e.g. with an inducible promoter or with cycloheximide) and determining whether TDP-43 returns to the soluble fraction or if it is degraded. A related question is whether autophagy or the proteasome are also required for clearance. Detailed answers to these questions would greatly boost the impact of this study and would represent an important advance to the field.

Reviewer #3 (Remarks to the Author):

This manuscript describes a follow-up study by Cohen and his colleagues on the possible role of acetylation in TDP-43 proteinopathies. In their 2015 Nature Communication paper, it was found that K145 acetylation of TDP-43 was detected in the pathogenic spinal cord samples from ALS (but not those from FTLD). This modification also appeared to enhance the TDP-43 aggregate formation in DNA transfection experiments in cell culture. In the current study, they carried out more experiments in cell cultures with use of acetylation mimic K~~Q~~ mutant TDP-43 within or without NLS. They also used a mouse skeletal muscle system trying to establish the in vivo role of K145 acetylation in TDP-43 proteinopathies.

Unfortunately, the study did not provide much new information regarding the effect of K145 acetylation on TDP-43 aggregation and its in vivo significance is unclear. Also, the neuroprotective effects of HSF1/ Hsp 27/ Hsp 40 in TDP-43 proteinopathies have been shown and reported by others already. Specific comments and questions are listed in the following:

- (1) The universal nature of K145 acetylation of TDP-43 in normal and diseased cells is unclear;
- (2) While the mouse skeletal muscle was used for overexpression experiments, there has been no evidence for the existence of K145 acetylation of TDP-43 in the skeletal muscles of patients with sporadic inclusion body myositis (sIBM);
- (3) The neuroprotective effects of HSF1/ Hsp 27/ Hsp 40 in TDP-43 proteinopathies have already been reported by other groups recently, eg. Shaw and his colleagues;
- (4) Following the above, there was a lack of time-dependence data in the study; also, why didn't the authors use motor neuron cultures, eg. NSC 34, for their DNA transfection/ overexpression experiments, if K145 acetylation was detected thus far only in ALS?
- (5) Only the K-to-Q mutant TDP-43 was used in the present study. To be more convincing with respect to the role of K145 acetylation, K-to-R mutant should also be used as a control for

comparison in most, if not all, of the experiments;

(6) Following the above, a number of experiments are also lack of controls, eg. Overexpression of wild type TDP-43 or TDP-43 (Δ NLS);

(7) The data in many figures are lack of statistical analysis/ presentation , eg. Fig. 1e, 2e, 6c, 6g, 8b, 8g, etc.

(8) The text is full of typo- and non-typo errors, eg. line 12 on p.11, line 17 on p.12, and many many others;

(9) Were phosphorylation, ubiquitination and acetylation occurring on the same TDP-43 molecule(s)?

Response to comments from Reviewers:

Below, we respond to the reviewer's comments and describe additional experiments and revisions that we have done to address their concerns. For your convenience, we highlight in yellow the key points in the comments that we addressed, and our responses to each of these points are highlighted in blue print.

Reviewer #1 (Remarks to the Author):

This group has previously found that TDP-43 can be acetylated at specific lysine residues. The present follow-up work initially confirms acetylation at K145 in the absence of ectopic acetyltransferase, a desired control experiment. At least for transfected TDP-43 protein with mutated nuclear localization sequence (Δ NLS) in QBI-293 cells. The authors go on with GFP fusion proteins to show in transfected QBI-293 cells an in electroporated muscle that supposedly acetylation mimicking K145Q substitutions enhance TDP-43 aggregation, particularly when combined with Δ NLS. Co-localization of the resulting inclusions with several components of the ubiquitin-proteasome system is taken as evidence for impaired protein clearance. Finally, hypothesis-driven examination of an involvement of HSF1 is tested, and systematic analysis of HSF1-dependent chaperones reveals the participation of 40kDa and 27kDa heat shock proteins.

The manuscript is well written and clear to follow, including the methods descriptions. Overall the results support the conclusions and the interpretations are reasonable. Although a follow-up on previously reported TDP-43 acetylation and aggregation, the present study provides novel detail information for the cognate chaperone defense program. The in vivo part is particularly relevant for the muscle TDPopathy, IBM.

Specific Comments:

1-1. More stringent language would use the term "acetylation-mimic" instead of "acetylated" when describing properties of the K->Q substituted TDP-43. Please check the manuscript text throughout.

We have now re-worded the text throughout to indicate, more accurately, that acetylation-mimicking TDP-43 mutations promote TDP-43 aggregation in this study. We note that our previous study used the acetyltransferase CBP to formally acetylate TDP-43 in cultured cells, which led to a very similar TDP-43 aggregation phenotype (Cohen et al., Nat Commun, 2015 Jan 5;6:5845).

1-2. The staining patterns of TDP-43 GFP fusion proteins in muscle should be described more carefully. First of all, endogenous TDP-43 staining in control tissue must be shown as a reference. Why does KQ-TDP-43-WT appear nuclear excluded, in stark contrast to the staining pattern in 293 cells? Why does the Δ NLS mutant show a staining pattern near nuclei and not distribute further out into the cytosol?

We have now included an immunohistochemical (IHC) stain showing endogenous TDP-43 staining in normal skeletal muscle. As expected, TDP-43 shows a predominantly nuclear distribution. This data is now included in Figure 5a.

Based on our staining patterns in 293 cells compared to muscle fibers, we believe that TDP-43

aggregation has notable cell-type specific differences. This is likely due to the fact that TDP-43 nucleo-cytoplasmic shuttling mechanisms may vary in large multi-nucleated myofibers compared to 293 cells, for example. Consistently, we observed the nuclear TDP-K145Q to be partially aggregated in the muscle sarcoplasm, as opposed to the punctate nuclear distribution that we typically observe in 293 cells (Figure 2). It is possible that TDP-K145Q is simply more aggregate-prone in skeletal muscle and therefore rapidly aggregates after being translated in the cytoplasm, then fails to be efficiently imported into the nucleus, resulting in more pronounced cytoplasmic aggregation in muscle. Similarly, the Δ NLS mutant may be more aggregate-prone in muscle fibers where it appears to preferentially accumulate adjacent to the nucleus (possibly adhered to the nucleus), reflecting a partially aggregated phenotype.

We have now carefully documented these intriguing localization differences in Figure 2a and Supplementary Figure 3 and elaborated in the results section to explain these differences (page 7, 1st paragraph). These data are consistent with important differences in nucleo-cytoplasmic TDP-43 shuttling *in vivo* in muscle, a feature that has not been analyzed before.

1-3. The quality of several dual-label immunostainings is insufficient.

a. **Fig. 2c phospho-TDP-43 immunostaining suffers from high background.** Much “red” is in areas not seen by GFP epifluorescence.

We agree that there is some background immunoreactivity using the commercially available phospho-TDP-43 antibody (from Proteintech). This is the most specific and well-characterized antibody that is currently used and available to detect TDP-43 pathology. We note that the phospho-TDP-43 inclusions are readily detectable above background, and most importantly, all of our double-labeling experiments have been confirmed independently by biochemical techniques (immunoblotting) that further validate these conclusions (Figure 2d).

b. Similar criticism for the ubiquitin co-stain, although here higher “background” signals may be expected. **Nevertheless, several ubiquitin “inclusions” are not mirrored in the green channel** (Fig. 3a).

We agree that the co-localization of TDP-43 with ubiquitin is not complete. We attribute this to cell-type specific regulation of TDP-43 aggregates. Not all inclusions appear to be ubiquitinated, which is consistent with a role for *both* ubiquitin-proteasome and autophagy pathways in mediating TDP-43 degradation, as we and others have previously suggested. In fact, in our revisions, we now show that both degradation pathways appear to target the acetylation-mimic TDP-43 aggregates (Supplementary Figure 9, and see reviewer comment 2-1).

c. In figure 4a there must be channel bleeding. **Or why would ubiquilin and optineurin become detectable only in transfected cells?**

We thank the reviewer for pointing out this discrepancy. Figure 4a has now been updated to clarify UBQLN-2 and OPTN recruitment to TDP-43 inclusions. We used a UBQLN-2 antibody to detect endogenous UBQLN-2 that is recruited to TDP-43 inclusions. Since OPTN antibodies were not adequate for similar endogenous OPTN analysis, an OPTN over-expression plasmid was co-transfected with TDP-43-GFP to enhance the detection sensitivity of OPTN and reduce non-specific background staining. These updates have now been described in the Figure 4 legend and the methods section has been updated to reflect these technical details.

d. Some longitudinal sections may be illustrative to see the nucleocytoplasmic distributions along the myofiber.

We have now provided a longitudinal muscle section depicting cytoplasmic acetylation-mimic TDP-43 inclusions that accumulate prominently along the muscle periphery and membrane, in contrast to the exclusively nuclear wild-type TDP-43. These new data are provided in the revised manuscript in Supplementary Figure 3b.

1-4. It is not quite clear what to make of the Western blot smears of mitochondrial marker proteins. Please show immunostainings. And refer to precedent literature, if applicable.

We now provide new immunostaining images showing mitochondria markers (Cytochrome c and COXIV) that co-localize strongly with TDP-43 inclusions in muscle (revised Figure 3d). Our data are consistent with TDP-43 aggregation leading to mitochondria aggregation, which has been previously observed in TDP-43 transgenic mice (for example, see Shan et. al, Proc Natl Acad Sci U S A. 2010 Sep. 16325-30 and Xu et al., J Neurosci. 2010 Aug 11;30(32):10851-9). In addition, we also observed mitochondrial aggregation as a high molecular weight smear in insoluble fractions. We now provide a reference that supports mitochondria aggregation as a potential biochemical phenomenon as well (page 9, line 2).

1-5. Please show controls in Fig. 5a,b. What is intended to demonstrate with Fig. 5d, right panel?

We have now added a non-electroporated control in Figure 5a that clearly illustrates endogenous TDP-43 localization side-by-side with our images showing acetylation-mimic TDP-43 inclusions in skeletal muscle (Figures 5a and 5b).

1-6. Supplementary Figure 5 and 6 are swapped.

We have now corrected this in the ordering of the supplementary figures (new Supplementary Figures 11 and 13).

1-7. Page 11, line 21: "...depletion of endogenous HSF1 did not significantly alter TDP-43 aggregation..." why so cautious here? One does get a visual impression of more aggregation (?)

We agree with the reviewer that HSF1 depletion showed a trend towards increased TDP-43 aggregation. However, our quantification of these results (based on number of aggregates and overall fluorescent intensity using automated software) did not show a significant difference under these conditions. Nonetheless, we have re-worded this statement to reflect a possible increased aggregation upon removal of HSF1.

On page 12, line 19, it now reads, "While siRNA-mediated depletion of endogenous HSF1 in myofibers showed a trend towards increased TDP-43 aggregation, this was not statistically significant under the parameters and conditions tested (Fig. 7d)".

1-8. Discussion: Acetylation of K145 within RRM1 could affect RNA binding, etc. Of course, this reasonable assumption would be better supported by experimental evidence.

Our previous study demonstrated that the K145Q acetylation-mimic showed ~ 50% reduction in TDP-43 binding to cellular mRNAs using a radiolabeled cross-linking IP (CLIP) assay. Please see Cohen et al. Nat Commun. 2015, 6:5845. Therefore, we now provide this reference in the introduction and discussion to further clarify this point.

Reviewer #2 (Remarks to the Author):

In this interesting study, Cohen and colleagues provide evidence consistent with acetylation-induced TDP-43 aggregation in cells and in vivo. This aggregation can be buffered and even partially reversed by introduction of activated HSF1, which elicits a transcriptional program of heat-shock proteins. Of these, Hsp27 and Hsp40 appear to be particularly important for antagonizing aggregation. Interestingly, it is possible to elicit these same effects pharmacologically using HSF1A a compound developed by the Thiele lab to induce a heat-shock response by engaging TriC. Overall, this study is interesting and deserves to be published, however a question remains that should be addressed prior to publication:

2-1. Evidence is presented suggesting that TDP-43 inclusions can be disaggregated by HSF1-induced chaperones. However, **what is the fate of the disaggregated TDP-43?** Is it refolded or does it get degraded? This question could be address by shutting down TDP-43 expression (e.g. with an inducible promoter or with cycloheximide) and determining whether TDP-43 returns to the soluble fraction or if it is degraded. A related question is whether autophagy or the proteasome are also required for clearance. Detailed answers to these questions would greatly boost the impact of this study and would represent an important advance to the field.

We now provide a new experiment to address this point. As shown in Supplementary Figure 9, sequential transfections were performed to allow HSF1 to disaggregate pre-formed TDP-43 aggregates. Subsequently, protein degradation was blocked using either the proteasome inhibitor MG-132 or the autophagy inhibitor 3-methyladenine (3-MA). Thus, using this approach suggested by the reviewer, we assessed whether disaggregated TDP-43 is targeted for degradation. Given the short ~ 48 hr duration of this experiment, general protein synthesis inhibitors were not technically compatible with optimal HSF1 expression and disaggregase activity due to the shortened timeframe, which we find did not impact the overall conclusions.

We found that blocking either pathway (proteasome or autophagy) was sufficient to restore the insoluble aggregated pool of TDP-43, but we observed a much more pronounced effect with MG132, which nearly eliminated the disaggregase activity of HSF1. Thus, it appears that HSF1 mediates TDP-43 disaggregation and then targets it for both proteasomal and autophagic degradation, with a more dominant role for the proteasome. If either pathway is perturbed, the insoluble pool of TDP-43 was increased (see Supplementary Figure 9b).

Secondly, we note that the soluble pool of TDP-43 is not significantly impacted by any chaperone manipulations (e.g. HSF1, Hsp27, or Hsp40 expression) or by proteasome/autophagy inhibition, suggesting it is the insoluble TDP-43 pool that is physically engaged and subject to a highly regulated refolding and degradation process.

Reviewer #3 (Remarks to the Author):

This manuscript describes a follow-up study by Cohen and his colleagues on the possible role of acetylation in TDP-43 proteinopathies. In their 2015 Nature Communication paper, it was found that K145 acetylation of TDP-43 was detected in the pathogenic spinal cord samples from ALS

(but not those from FTLD). This modification also appeared to enhance the TDP-43 aggregate formation in DNA transfection experiments in cell culture. In the current study, they carried out more experiments in cell cultures with use of acetylation mimic K145Q mutant TDP-43 within or without NLS. They also used a mouse skeletal muscle system trying to establish the in vivo role of K145 acetylation in TDP-43 proteinopathies.

Unfortunately, the study did not provide much new information regarding the effect of K145 acetylation on TDP-43 aggregation and its in vivo significance is unclear. Also, the neuroprotective effects of HSF1/ Hsp 27/ Hsp 40 in TDP-43 proteinopathies have been shown and reported by others already. Specific comments and questions are listed in the following:

3-1 **The universal nature of K145 acetylation of TDP-43 in normal and diseased cells is unclear;**

Using histological staining methods with an acetylation-specific TDP-43 antibody, we previously detected acetylated TDP-43 inclusions in ALS spinal cord but not controls (please see Cohen et al. Nat Commun. 2015, 6:5845). Thus, we considered that TDP-43 is hyper-acetylated in some TDP-43 proteinopathies, but this does not exclude the possibility that normal control tissues maintain low levels of acetylated TDP-43. This latter possibility is consistent with basal acetylation of another pathological protein, the microtubule-associated tau protein, which we and others have shown becomes hyper-acetylated in Alzheimer's disease but is also present in normal control brains at low basal levels. We note that our results in skeletal provide a conceptual advance for the role of TDP-43 acetylation in triggering its aggregation in vivo, which has been very challenging to study in other systems. Thus, while we agree with the reviewer that it still unclear exactly how TDP-43 acetylation is linked to disease, our study provides new support that this modification is clearly linked to abnormal accumulation of TDP-43 pathology.

3-2 While the mouse skeletal muscle was used for overexpression experiments, **there has been no evidence for the existence of K145 acetylation of TDP-43 in the skeletal muscles of patients with sporadic inclusion body myositis (sIBM);**

We agree with the reviewer that the link between TDP-43 acetylation and sIBM is currently unknown. However, as mentioned above in comment 3-1, we showed previously that acetylated TDP-43 is present in ALS patient spinal cord comprised of full-length TDP-43 pathology, which certainly warrants investigation into the role of TDP-43 acetylation in sIBM patient tissue. This is an exciting future direction that could fill a major gap in our understanding of how TDP-43 pathology evolves in distinct tissue-types (e.g. brain, spinal cord, and muscle) to produce distinct clinical outcomes. Nonetheless, our ability to generate TDP-43 pathology "de novo" using acetylation-mimic mutations provides an excellent starting point to further understand how sIBM pathology emerges. We certainly expect that our studies using multiple in vivo models will address how TDP-43 pathology evolves in different tissue types.

3-3 The neuroprotective effects of HSF1/ Hsp 27/ Hsp 40 in TDP-43 proteinopathies have already been reported by other groups recently, eg. Shaw and his colleagues;

We agree that a previous group reported that HSF-1 alters TDP-43 solubility in cultured cells (Chen et al. Brain. 2016, 5, 1417-32). Indeed, we were careful to reference this study both in the introduction and the discussion sections. However, we note that our results are the first to investigate HSF-1 regulation of TDP-43 in vivo (in skeletal muscle) and also to demonstrate the necessity for Hsp27 and Hsp40 using HSP siRNA loss of function approaches. Honing in on the relevant HSPs downstream of HSF1 has major implications for clearing TDP-43 aggregates in

muscle, a tissue type that forms robust TDP-43 pathology and undergoes degeneration in sIBM patients. Our results show that Hsp27 and Hsp40, in particular, have protective roles in suppressing TDP-43 pathology in cells and muscle fibers; this overall conclusion has not been shown in prior studies (Figures 8 and 9).

3-4 Following the above, there was a lack of time-dependence data in the study; also, **why didn't the authors use motor neuron cultures, eg. NSC 34,** for their DNA transfection/ overexpression experiments, if K145 acetylation was detected thus far only in ALS?

Based on the reviewer's comment, we have now examined TDP-43 acetylation-mimic mutants in NSC-34 cells. We provide new data showing that the K145Q mutation leads to TDP-43 aggregation in NSC-34 cells, which is included in the revised Supplementary Figure 2. The timing of all of our cell-based transfections or muscle electroporations is now clearly described in each figure legend or in the methods sections.

3-5 Only the K-to-Q mutant TDP-43 was used in the present study. To be more convincing with respect to the role of K145 acetylation, **K-to-R mutant should also be used as a control for comparison in most, if not all, of the experiments;**

We agree with the reviewer that a critical control mutant is the K→R mutation. In our previous study, we analyzed all possible K→R and K→Q mutations (at several lysine residues, both K145 and the related K192). We assessed these mutants using multiple approaches; immunofluorescence, biochemical experiments, and functional assays (see Cohen et al. Nat Commun. 2015, 6:5845). *In that study, we showed that either lysine acetylation (mediated by the acetyltransferase CBP/p300) or K145Q acetylation-mimics, but not K145R non-mimic mutations, were sufficient to promote TDP-43 aggregation and loss of TDP-43 RNA binding function.* We have now clarified this on page 15, line 15, which now reads, "Our prior study suggested that acetylation achieved with the acetyltransferase CBP or K→Q mimics, but not K→R non-mimics, could abrogate TDP-43-RNA interactions leading to an unstable TDP-43 conformation and aggregation³⁰."

Secondly, we now also include a K→R mutant control in our NSC-34 analysis, which did not show any detectable aggregation (Supplementary Figure 2).

3-6 Following the above, **a number of experiments are also lack of controls, eg. Overexpression of wild type TDP-43 or TDP-43 (ΔNLS);**

We showed that wild-type TDP-43 was exclusively nuclear in cells (Figure 1) or muscle fibers (Figure 2), in agreement with many previous studies. Only K145Q mutants showed a robust aggregation phenotype, as shown in Figures 2a and 2b, and therefore we focused our efforts on characterizing the mutant pathology throughout this paper. We now include additional control images showing *1) nuclear localization of endogenous TDP-43 in muscle fibers (Figure 5a), 2) nuclear localization of electroporated wild-type TDP-43 in muscle fibers (Figure 5b), and 3) longitudinal muscle sections that confirm the nuclear pattern of wild-type TDP-43, while acetylation-mimic TDP-43 was aggregated at the muscle membrane (Supplementary Figure 3b).*

3-7 **The data in many figures are lack of statistical analysis/ presentation , eg. Fig. 1e, 2e, 6c, 6g, 8b, 8g, etc.**

We have now updated all of the necessary panels in Figs 1-9 in the paper with statistical analysis. Please see the revised Figs 1e, 2e, 6c, 6e, 6g, 6i, 7d, 8b, 8f, 8g, 9b. Significance tests

are clearly indicated in the figure legends with appropriate p-value designations (*, **, or, *** equals p-value < 0.05, 0.01, or 0.001).

3-8 The text is full of typo- and non-typo errors, eg. line 12 on p.11, line 17 on p.12, and many many others;

All grammatical and formatting errors have been corrected throughout the paper. We thank the reviewer for pointing this out.

3-9 Were phosphorylation, ubiquitination and acetylation occurring on the same TDP-43 molecule(s)?

The stoichiometry of these different modifications per molecule of TDP-43 is currently unclear, partly due to the limited resolution at the single molecule level that can be achieved with our acetylation analysis *in vivo*. However, as the reviewer points out, TDP-43 can undergo different modifications (e.g. acetylation at K145 and K192, phosphorylation at S403, S404, S409, S410, and ubiquitination). A more detailed mass spectrometry study of isolated TDP-43 from ALS tissue is currently underway to assess whether any TDP-43 peptides identified contain multiple modifications on neighboring or adjacent residues. Those studies are currently ongoing and will form the basis of an exciting future study that is beyond the scope of the current manuscript.

Reviewers' comments:

Reviewer #1 (Remarks to the Author):

The authors have responded well to the reviewer criticisms. I think the manuscript is ready for publication with one last, minor amendment (original concern 1-3b): as there is consensus that "co-localization of TDP-43 with ubiquitin is not complete", the authors should state so in the results text, e.g. page 8, line 163 "...>some< cytoplasmic aggregates were ubiquitinated...".

Please also correct typo on page 7, line 138: Confocal imaging not "imagining" :-)

Reviewer #2 (Remarks to the Author):

The authors have addressed my previous concerns and the paper is ready for publication.

Reviewer #3 (Remarks to the Author):

In this revised manuscript by Wang et al., the authors have added new data and make changes to suggest a link between acetylation of TDP-43 and possible TDP-43 proteinopathies in the sporadic inclusion body myositis (sIBM), a human disease with pathology in the skeletal muscle. This suggestion was achieved by study of the effects of overexpression of an artificial acetylation-mimic polypeptide TDP-43- Δ NLS-KQ in muscle cells. Further, from their data, the authors suggested that HSF1 along with the induced Hsp40 and Hsp27 would reduce the aggregation/ inclusion formation upon overexpression of the mutant TDP-43- Δ NLS-KQ.

I still have reservations and major concerns about the study. The amount of the data provided is huge, but some essential ones are problematic. Key experiments are also lacking. My comments are as follows:

(1) Lack of biological significance and novelty of the findings. The possible links among TDP-43 acetylation, TDP-43 proteinopathies, and sIBM are merely speculation. The formation of the cytotoxic aggregates by the mutant protein is expected to occur in many types of cells, and the authors have already shown for several different cell types in their previous publication in Nat. Comm.,

(2) The data attempting to support the fundamental role of HSF1, Hsp40 and Hsp 27 in reducing the aggregation of TDP-43- Δ NLS-KQ did not convince this reviewer:

(i) Despite of the authors' response to reviewer #1, I still don't think that data in Fig. 7 support the authors' conclusion;

(ii) The experiments presented in Fig. 9 and Supplementary Fig. 13 are lack of essential controls. Specifically, there were no data from (Δ NLS-KQ + Hsp40 siRNA), (Δ NLS-KQ + Hsp27 siRNA), (Δ NLS-KQ + HSF1+control siRNA);

(3) The data and relevant description of Supplementary Fig. 7 are puzzling -

First, the figure panels were labeled as "TDP-43- Δ NLS-K145Q", but the text claimed that the cells were overexpressing "TDP-43"; Second, did the authors compare the scores of EBD uptake between cells overexpressing wild type TDP-43, TDP-43- Δ NLS, and TDP-43- Δ NLS-KQ, respectively? Thirdly, the cell with EBD uptake shown in the figure had no TDP-43? Also, statistics?

(4) In Fig. 1d, the blot containing the insoluble fraction of TDP-43 was not "cropped" from the "original blot" displayed in Supplementary Fig. 1, as claimed by the authors.

The PI and editor may want to double check whether this was merely caused by incident or due to an ethics problem;

(5) Problems of Supplementary Fig. 2.

(i) Why were there blue signals in most of the panels?

(ii) Why were the distribution patterns of "TDP-43- Δ NLS-K145R" and "TDP-43- Δ NLS" different?

(6) In Fig. 5, only the images from "TDP-43" were compared to those from "TDP-43- Δ NLS-K145Q" on day 22. The appropriate control to use should be "TDP-43- Δ NLS".

(7) The manuscript still contains numerous typo errors and/or mis-citing of the reference numbers. Further, there is still a lack of statistical analysis in some of the figures.

2nd Revision: Response to Comments from Reviewers:

Below, we respond to the reviewer's comments and describe additional experiments and revisions that we have done to address their concerns. For your convenience, **we highlight in yellow the key points in the comments that we addressed**, and our responses to each of these points are highlighted in blue print.

(1) **Lack of biological significance and novelty of the findings**. The possible links among TDP-43 acetylation, TDP-43 proteinopathies, and sIBM are merely speculation. The formation of the cytotoxic aggregates by the mutant protein t is expected to occur in many types of cells, and the authors have already shown for several different cell types in their previous publication in Nat. Comm.,

As mentioned in the first round rebuttal letter, **this is the first study to show widespread and robust TDP-43 pathology in vivo (Figs 2,5,7)**. Experts generally agree that it is very challenging to generate TDP-43 pathology *in vivo* resembling the robust pathology seen in ALS or FTD patients. Our ability to generate such pathology stems from our important discovery of TDP-43 acetylation, which we now show is a striking promoter of TDP-43 aggregation.

(2) The data attempting to support the fundamental role of HSF1, Hsp40 and Hsp 27 in reducing the aggregation of TDP-43-ΔNLS-KQ did not convince this reviewer:

We show that HSF-1, Hsp27, and Hsp40 are all sufficient to dramatically suppress the insoluble aggregated pool of TDP by western and by immunohistochemical (IHC) techniques in cultured cells and muscle fibers (Figs 6-9, in some cases we achieved ~90% aggregate suppression). Biochemical and staining approaches are the two most well accepted strategies that define TDP-43 pathology. In addition, we performed siRNA experiments for hsp27 and 40 to show their necessity, which has not been shown in prior studies (Fig. 9). All of the pertinent controls and statistical analysis have been thoroughly documented in these experiments.

(i) **Despite of the authors' response to reviewer #1, I still don't think that data in Fig. 7 support the authors' conclusion;**

We clearly demonstrated in skeletal muscle that HSF-1 is protective against TDP-43 pathology and further quantified the TDP-43 pathology in the presence or absence of HSF-1 (either active or inactive HSF-1). There are no additional controls that are lacking from this experiment.

(ii) **The experiments presented in Fig. 9 and Supplementary Fig. 13 are lack of essential controls. Specifically, there were no data from (ΔNLS-KQ + Hsp40 siRNA), (ΔNLS-KQ + Hsp27 siRNA), (□NLS-KQ + HSF1+control siRNA);**

As described in the results section, the goal of this experiment was not to determine the importance of individual HSPs in regulating TDP-43 per se. **Rather, our goal was to determine which HSPs downstream of HSF-1 transcriptional activity are important in suppressing TDP-43 aggregation.** Therefore, we required HSF-1 to be present in the system (as a disaggregase) in order to determine the HSPs that are necessary for optimal HSF-1 disaggregase function. Indeed, our results clearly show that Hsp27 and Hsp40 are two of the critical chaperones that suppress TDP-43 aggregation (Figs 8 and 9).

(3) The data and relevant description of Supplementary Fig. 7 are puzzling-the figure panels were labeled as "TDP-43-ΔNLS-K145Q", but the text claimed that the cells were overexpressing "TDP-43";

This has been corrected and re-worded as "TDP-43-ΔNLS-K145Q".

Second, did the authors compare the scores of EBD uptake between cells overexpressing wild type TDP-43, TDP-43-ΔNLS, and TDP-43-ΔNLS-KQ, respectively? Thirdly, the cell with EBD uptake shown in the figure had no TDP-43? Also, statistics?

To clarify the rationale for this experiment, our goal was to determine whether the robust TDP-43 pathology achieved with the acetylation-mimic mutants causes membrane damage, as visualized by Evans Blue Dye (EBD) uptake. EBD is an injectable dye that is permeable only to damaged muscle fibers. It did not under any conditions penetrate muscle fibers containing large TDP-43 aggregates. **Thus, in contrast to that suggested by the reviewer, our EBD results show negative data, i.e. there is no apparent loss in muscle membrane integrity whatsoever in the presence of large TDP-43 inclusions, and regardless of the TDP-43 construct used.** We only occasionally noticed rare fibers that were EBD-positive, which we clearly marked by arrows.

(4) In Fig. 1d, the blot containing the insoluble fraction of TDP-43 was not “cropped” from the “original blot” displayed in Supplementary Fig. 1, as claimed by the authors. The PI and editor may want to double check whether this was merely caused by incident or due to an ethics problem;

We thank the reviewer for pointing this out. We performed this experiment with many (n=6) biological replicates, each with n=3 technical replicates per experiment. All replicated blots were nearly identical. We inadvertently used a full-length blot that did not match the cropped blot included in Fig. 1. We have replaced the exact version of the full-length blot in the revised Supplementary Fig.1.

(5) Problems of Supplementary Fig. 2.

(i) Why were there blue signals in most of the panels?

Including a blue DAPI signal in the panels depicting the merged immunostaining analysis is a standard technique used to mark nuclei and orient the readers to where nuclei and cytoplasm are located. We fully explained the use of DAPI in all relevant figure legends.

(ii) Why were the distribution patterns of “TDP-43-ΔNLS-K145R” and “TDP-43-ΔNLS” different?

Cytoplasmic targeted TDP-43-ΔNLS can show a subtle accumulation of cytoplasmic foci, which has been observed in many prior studies (e.g. see Winton et al. JBC, 2008). However, it is not highly aggregated. We detailed this phenomenon extensively using biochemical and imaging methods in Figs 1 and 2. In contrast, TDP-43 acetylation-mimic K→Q mutants show much more dramatic aggregation. As a control, we did not observe a dramatically different phenotype with the K→R non-mimic mutant, TDP-43-ΔNLS-K145R, as compared to TDP-43-ΔNLS in NSC-34 cells; both are simply cytoplasmic with some subtle accumulation of dispersed foci. Thus, the K→R non-mimic mutant was the appropriate control for this experiment and did not show any notable differences.

(6) In Fig. 5, only the images from “TDP-43” were compared to those from “TDP-43-ΔNLS-K145Q” on day 22. The appropriate control to use should be “TDP-43-ΔNLS”.

In Figs 1,2, and 5 (and Sup. Fig. 3) we clearly showed that K→Q mutants, but not wild-type or TDP-43-ΔNLS, are highly aggregate-prone, more so than any other TDP-43 modifications or disease-causing mutants analyzed previously. We convincingly demonstrated that only K→Q acetylation-mimic mutants generate the robust, mature, phospho-TDP-43-positive pathology in cells and muscle fibers. Since all other TDP-43 constructs other than K→Q mutants are minimally aggregated in muscles and do not reflect the “diseased sIBM phenotype”, we focused our efforts on defining the biochemical and histological properties of this intriguing K→Q mutant. The K→Q mutant, as far as we can tell, is the single most dominant controller of TDP-43 aggregation that we have worked on to

date. Since all control electroporations were established in earlier figures in our manuscript (Figs 1,2,5), documenting them again in Fig. 5 was not necessary.

(7) The manuscript still contains numerous typo errors and/or mis-citing of the reference numbers. Further, there is still a lack of statistical analysis in some of the figures.

The manuscript is now meticulously written; there are no other grammatical or typo errors, to the best of our knowledge. Reviewer #1 correctly pointed out one additional typo that we have now fixed.